# A dynamic mechanism for allosteric activation of Aurora kinase A by activation loop phosphorylation

**Emily F Ruff[1]†, Joseph M Muretta[2], Andrew R Thompson[2], Eric W Lake[1], Soreen Cyphers[1], Steven K Albanese[3,4], Sonya M Hanson[3], Julie M Behr[3,5], David D Thomas[2], John D Chodera[3], Nicholas M Levinson[1]***

[1]Department of Pharmacology, University of Minnesota, Minneapolis, United States; [2]Department of Biochemistry, Molecular Biology, and Biophysics, University of Minnesota, Minneapolis, United States; [3]Computational and Systems Biology Program, Sloan Kettering Institute, Memorial Sloan Kettering Cancer Center, New York, United States; [4]Gerstner Sloan Kettering Graduate School, Memorial Sloan Kettering Cancer Center, New York, United States; [5]Tri-Institutional Program in Computational Biology and Medicine, Weill Cornell Medical College, New York, United States

**\*For correspondence:**
nml@umn.edu

**Present address:** †Department of Chemistry, Winona State University, Winona, United States

**Abstract** Many eukaryotic protein kinases are activated by phosphorylation on a specific conserved residue in the regulatory activation loop, a post-translational modification thought to stabilize the active DFG-In state of the catalytic domain. Here we use a battery of spectroscopic methods that track different catalytic elements of the kinase domain to show that the ~100 fold activation of the mitotic kinase Aurora A (AurA) by phosphorylation occurs without a population shift from the DFG-Out to the DFG-In state, and that the activation loop of the activated kinase remains highly dynamic. Instead, molecular dynamics simulations and electron paramagnetic resonance experiments show that phosphorylation triggers a switch within the DFG-In subpopulation from an autoinhibited DFG-In substate to an active DFG-In substate, leading to catalytic activation. This mechanism raises new questions about the functional role of the DFG-Out state in protein kinases.
DOI: https://doi.org/10.7554/eLife.32766.001

## Introduction

Stringent regulatory control of protein kinases is critically important for the integrity of cellular signal transduction. The catalytic activity of protein kinases is regulated by finely-tuned allosteric mechanisms that reversibly switch the kinase domain between active and inactive conformational states (*Huse and Kuriyan, 2002*). Disruption of these mechanisms, leading to constitutive kinase activity, is a major cause of human cancer, and small molecules that inhibit specific disease-associated kinases are an increasingly important component of many modern cancer therapies (*Zhang et al., 2009*).

Phosphorylation on a specific site in the activation loop of the kinase domain is the most widely conserved regulatory mechanism in kinases (*Johnson et al., 1996*). X-ray structures show that ionic interactions between the phosphate moiety and a surrounding pocket of basic residues stabilize the activation loop in a conserved active conformation (*Knighton et al., 1991*; *Yamaguchi and Hendrickson, 1996*; *Steichen et al., 2012*). In this active state, a catalytic asp-phe-gly (DFG) motif at the N-terminal end of the activation loop adopts an active 'DFG-In' conformation, with the aspartate residue of the DFG motif pointing into the active site to coordinate Mg-ATP, and the C-terminal segment of the activation loop positioned to bind peptide substrates. The DFG-In state is stabilized by

**eLife digest** The transfer of phosphate groups onto proteins (protein phosphorylation) is one of the most important methods used to send signals inside cells. The enzymes that catalyze this process, called protein kinases, are themselves controlled by the phosphorylation of a flexible region called the activation loop.

For many years it had been thought that the purpose of activation loop phosphorylation was to clamp the otherwise flexible activation loop in an active state that allows molecules that need to be phosphorylated to bind to the kinase. This assumption was based on static pictures of protein kinases obtained by X-ray crystallography, in which individual states are trapped and visualized in a crystal lattice. However, new methods and approaches now mean it is possible to visualize how the position of the activation loop changes as it moves in solution. By applying these techniques, Ruff et al. show that the static model is incorrect in a protein kinase called Aurora A. In this enzyme, the phosphorylated activation loop continues to switch back and forth between active and inactive states. Phosphorylation instead enhances the catalytic activity of the active state.

Aurora A regulates several important steps in cell division, and plays important roles in several kinds of cancer. The discovery that activated forms of Aurora A can have different dynamic properties raises the possibility that inhibitor molecules could be designed to exploit these differences and block specific activities of Aurora A in cancer cells. To realize this goal we need to better understand how a kinase switching between active and inactive states affects the ability of inhibitors to interact with it.

DOI: https://doi.org/10.7554/eLife.32766.002

the assembly of a network of hydrophobic residues, termed the regulatory spine, that lock together the N-terminal lobe, the αC-helix, the DFG motif and the C-terminal lobe of the kinase (*Kornev et al., 2006*; *Kornev et al., 2008*). In the absence of phosphorylation on the activation loop, kinase activity is usually restrained by rearrangements of the activation loop and DFG motif into specific autoinhibited conformations. An important autoinhibited state, called 'DFG-Out', involves a flip in the backbone torsion angles of the DFG motif, reorienting the DFG aspartate out of the active site to prevent magnesium coordination, repositioning the activation loop to block pep-tide binding, and disassembling the regulatory spine (*Hubbard et al., 1994*; *Nagar et al., 2003*; *Mol et al., 2004*). This conformational change dramatically alters the chemical makeup of the active site, and selective recognition of the DFG-Out state underlies the mechanism of action of many small-molecule kinase inhibitors (*Liu and Gray, 2006*).

The serine/threonine kinase Aurora A (AurA) is an essential mitotic protein that controls a variety of cellular processes including mitotic spindle assembly, centrosome maturation, and mitotic entry (*Glover et al., 1995*; *Hannak et al., 2001*; *Berdnik and Knoblich, 2002*; *Macůrek et al., 2008*; *Seki et al., 2008*). These functions of AurA are driven by two distinct activation mechanisms of the kinase operating in different spatiotemporal contexts. At the centrosome, AurA must first be acti-vated by autophosphorylation on the activation loop threonine residue T288 in order to carry out its centrosomal functions. At the mitotic spindle, AurA is instead activated by binding to the spindle assembly factor Tpx2 (*Kufer et al., 2002*). This spindle-associated pool of AurA must be maintained in the unphosphorylated state by the phosphatase PP6 in order for spindle assembly to proceed faithfully (*Zeng et al., 2010*; *Toya et al., 2011*). Extensive in vitro studies have confirmed that Tpx2 and phosphorylation can act independently to increase AurA kinase activity by up to several hun-dred-fold (*Zorba et al., 2014*; *Dodson and Bayliss, 2012*).

We recently showed that activation of AurA by Tpx2 is driven by a population shift from a DFG-Out to the DFG-In state (*Cyphers et al., 2017*). Since crystal structures of phosphorylated AurA bound to Tpx2 show the T288 phosphothreonine residue forming the canonical ionic interactions thought to stabilize the DFG-In state (*Bayliss et al., 2003*; *Zhao et al., 2008*; *Clark et al., 2009*), it has been assumed that phosphorylation also triggers a transition from the DFG-Out to the DFG-In state. In this paper, we show that phosphorylation on T288 in fact activates AurA through a completely different mechanism than Tpx2. Three complementary spectroscopic methods, infrared spectroscopy, Förster resonance energy transfer, and double electron-electron resonance, all show

that phosphorylation does not trigger a switch to the DFG-In state, and that phosphorylated AurA continually samples both DFG-In and DFG-Out conformational states. Instead, phosphorylation triggers a conformational switch from a previously unknown inactive DFG-In substate to a fully activated DFG-In substate, enhancing the catalytic activity of the DFG-In subpopulation within a dynamic conformational ensemble.

## Results

### Phosphorylation of AurA on T288 does not switch the kinase into the DFG-In state

We set out to explain how phosphorylation of AurA on T288 leads to a ~100 fold increase in catalytic activity (*Figure 1—figure supplement 1a*) (*Zorba et al., 2014*; *Dodson and Bayliss, 2012*). We previously used an infrared (IR) probe that tracks the DFG motif of AurA to show that Tpx2 binding triggers a conformational change from the DFG-Out to the DFG-In state (*Cyphers et al., 2017*), resulting in the assembly of the active site and the regulatory spine (*Figure 1—figure supplement 2*). In this method, a cysteine residue is introduced at position Q185 at the back of the active site of AurA, and chemical labeling is used to introduce a nitrile infrared probe at this position (*Fafarman et al., 2006*). To test whether phosphorylation of AurA also causes a conformational shift of the DFG motif, we prepared samples of AurA Q185C phosphorylated on T288. Homogeneous phosphorylation and nitrile labeling were verified by western blotting and mass spectrometry (*Figure 1—figure supplement 1*).

IR spectra of nitrile-labeled phosphorylated AurA showed predominantly a single absorbance band centered at 2158 $cm^{-1}$ (*Figure 1a*, solid black line). We previously assigned this peak in IR spectra of unphosphorylated AurA to the DFG-Out form of the kinase, in which the nitrile probe is buried in a hydrophobic pocket (*Figure 1b*, lower panel) (*Cyphers et al., 2017*). Addition of saturating amounts of Tpx2 peptide (residues 1–43 of human Tpx2) to the IR samples caused a dramatic spectral change wherein the central peak at 2158 $cm^{-1}$ is largely replaced by two new peaks at 2149 $cm^{-1}$ and 2164 $cm^{-1}$ (*Figure 1a*, dashed black line). These changes are indicative of a shift to the DFG-In state, in which water molecules coordinated to the DFG motif form hydrogen bonds to the nitrile probe, causing pronounced spectral shifts (*Figure 1b*, upper right panel) (*Cyphers et al., 2017*). To confirm that the peak at 2158 $cm^{-1}$ arises from the DFG-Out state, we mutated residue W277, which is positioned directly against the IR probe in the DFG-Out state, but is displaced away from it in the DFG-in state, to alanine (*Figure 1b*). IR spectra of the W277A mutant showed a clear spectral shift of the 2158 $cm^{-1}$ peak (*Figure 1c*), consistent with this peak arising from the DFG-Out state.

The addition of ADP to apo AurA resulted in the appearance of a DFG-In subpopulation, apparent in the IR spectra as small shoulders on either side of the main 2158 $cm^{-1}$ peak. Experiments performed over a range of temperatures showed that this DFG-In subpopulation increases at higher temperature (*Figure 1a*, colored lines), but does not reach the level observed in the presence of Tpx2. A similar DFG-In subpopulation was also detected in unphosphorylated AurA bound to ADP (*Cyphers et al., 2017*) (*Figure 1a*, inset), highlighting that although nucleotide binding shifts the DFG equilibrium towards the DFG-In state, phosphorylation does not seem to enhance this effect. These IR results suggested that phosphorylation alone does not substantially change the DFG-In/Out equilibrium of AurA, unlike Tpx2 binding. However, as replacement of the Q185 residue with the nitrile probe was found to alter the activation properties of AurA (*Cyphers et al., 2017*), it was necessary to confirm this interpretation using an alternative method.

### The phosphorylated activation loop adopts a range of conformations in solution, and only becomes highly ordered upon Tpx2 binding

We used intramolecular FRET to track movements of the activation loop of AurA with and without phosphorylation on T288, using a construct with a native Q185 residue (*Cyphers et al., 2017*). Donor (D) and acceptor (A) fluorophores (Alexa 488 and Alexa 568, respectively) were incorporated on the activation loop (S284C) and αD helix (L225C) using maleimide chemistry (*Cyphers et al., 2017*). These labeling positions were chosen to track the movement of the activation loop across the active-site cleft as the kinase switches from the DFG-Out to the DFG-In state, with the dyes

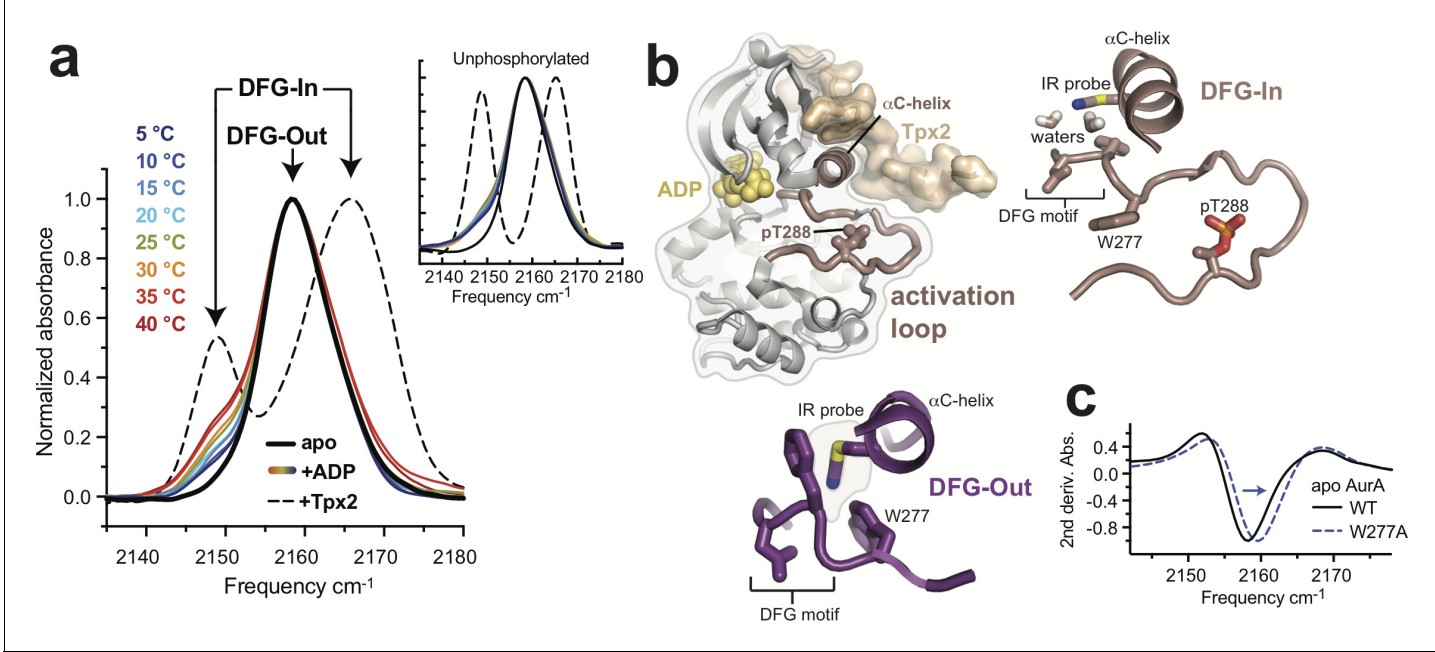

**Figure 1.** Phosphorylation on T288 does not switch AurA into the DFG-In state. (**a**) IR spectra of nitrile-labeled phosphorylated AurA. The apo sample (solid black line), and the sample bound to Tpx2 (dashed black line), were measured at 5°C, and the kinase bound to ADP (colored lines) was measured at the indicated temperatures. Arrows indicate peaks assigned to the DFG-In and DFG-Out states. The inset shows the same experiments performed with unphosphorylated AurA. Single representative spectra are shown, normalized to peak maxima. (**b**) Overview of the structure of AurA in the active conformation bound to ADP (yellow) and Tpx2 (beige), with enlarged views of the DFG-In (right, PDB ID: 1OL5) and DFG-Out (bottom, PDB ID: 5L8K) states with the nitrile probe (Q185CN) modeled into the structures. (**c**) Second derivatives of IR spectra of apo WT and W277A AurA, showing the ~1.5 cm$^{-1}$ spectral shift of the 2158 cm$^{-1}$ peak (arrow).

DOI: https://doi.org/10.7554/eLife.32766.003

The following figure supplements are available for figure 1:

**Figure supplement 1.** Activation of AurA by phosphorylation and preparation of homogenously phosphorylated nitrile-labeled AurA for IR experiments.

DOI: https://doi.org/10.7554/eLife.32766.004

**Figure supplement 2.** Comparison of the active site of AurA in the DFG-In and DFG-Out states.

DOI: https://doi.org/10.7554/eLife.32766.005

predicted to be further apart in the DFG-In state (*Figure 2a*). Phosphorylation of the protein on T288 was confirmed by tryptic mass spectrometry (*Figure 2—figure supplement 1*), and the labeled phosphorylated sample exhibited robust catalytic activity in the absence of Tpx2, and was further activated ~4 fold by the addition of Tpx2 (*Figure 2—figure supplement 1c*) (*Zorba et al., 2014*; *Dodson and Bayliss, 2012*).

Steady-state fluorescence emission spectra were measured for D- and D + A labeled forms of both unphosphorylated and phosphorylated AurA. In either phosphorylation state, titrating ADP or Tpx2 onto the kinase resulted in enhanced fluorescence emission from the donor dye and reduced emission from the acceptor, indicating a decrease in FRET efficiency (*Figure 2—figure supplement 2*) consistent with a shift towards the DFG-In state. To gain more insight into the conformation of the activation loop and how it is altered by phosphorylation and ligand binding, we performed time-resolved (TR) FRET experiments to quantify energy transfer through its effect on the fluorescence lifetime of the donor dye. TR fluorescence decays were recorded using time-correlated single-photon counting (TCSPC) (*Figure 2b*, top panel), and were then fit to a structural model consisting of a Gaussian distribution of inter-fluorophore distances (*Muretta et al., 2013*; *Agafonov et al., 2009*; *Nesmelov et al., 2011*) to represent the ensemble of conformations sampled in solution (*Figure 2b*, bottom panels).

The distance distributions measured for the phosphorylated and unphosphorylated kinase in the absence of ligands are strikingly similar (*Figure 2b*, bottom panels). In both cases, a broad

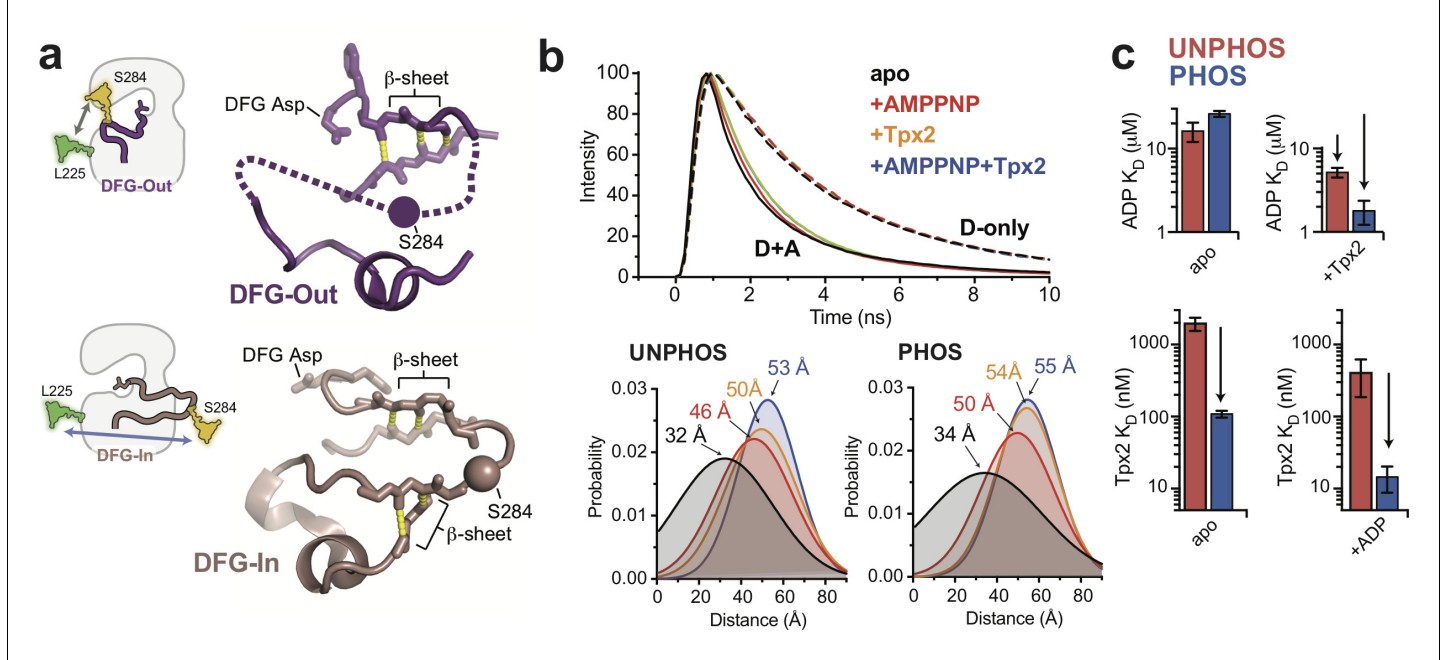

**Figure 2.** The phosphorylated activation loop remains flexible and shifts to a more active conformation upon Tpx2 binding. (a) (left) Schematics showing the labeling scheme used to detect the DFG-In/Out transition by FRET. (right) Structures of the DFG-Out (top) and DFG-In (bottom) states of AurA, highlighting the β-sheet hydrogen bonds constraining the N- and C-terminal segments of the activation loop. The S284C labeling site is shown as a sphere. (b) (top) Time-resolved fluorescence waveforms for D-only (dashed lines) and D + A (solid lines) phosphorylated AurA in the presence and absence of 125 µM Tpx2 and 1 mM AMPPNP. Data are for a single representative experiment, normalized to the fluorescence peak. (bottom) Comparison of single-Gaussian distance distribution fits to fluorescence lifetime data obtained with unphosphorylated (left) and phosphorylated AurA (right). (c) Binding constants of ADP (top panels) and Tpx2 (bottom panels) for phosphorylated (blue) and unphosphorylated (red) AurA determined with and without the other ligand pre-bound to the kinase. Data represent mean values ± s.d.; n = 3.
DOI: https://doi.org/10.7554/eLife.32766.006

The following figure supplements are available for figure 2:

**Figure supplement 1.** Validation of constructs used for fluorescence experiments.
DOI: https://doi.org/10.7554/eLife.32766.007

**Figure supplement 2.** Steady-state fluorescence and FRET experiments.
DOI: https://doi.org/10.7554/eLife.32766.008

distribution centered around ~30 angstroms is observed for apo AurA, indicating that the activation loop is highly flexible regardless of the phosphorylation state (*Figure 2b*, black). This broad distribution is consistent with the DFG-Out state, in which the C-terminal half of the activation loop lacks contacts with the rest of the kinase domain, and is typically disordered in X-ray structures (*Wu et al., 2013*; *Coumar et al., 2009*; *Fancelli et al., 2006*) (*Figure 2a*, top panel). In contrast, when the phosphorylated and unphosphorylated samples were saturated with both Tpx2 peptide and nucleotide (either ADP or the non-hydrolysable ATP-analog AMPPNP), narrow distributions were observed that were shifted to ~54 angstroms (*Figure 2b*, blue). This indicates adoption of a well-defined structure consistent with the DFG-In state, in which the segment of the loop containing the labeling site is anchored to the C-terminal lobe of the kinase by flanking β-sheet interactions (*Bayliss et al., 2003*) (*Figure 2a*, bottom panel). In the presence of ADP or AMPPNP alone the observed distance distributions were intermediate in both distance and width between the other samples, consistent with nucleotide binding driving unphosphorylated and phosphorylated AurA into a similar equilibrium between DFG-Out and DFG-In states (*Figure 2b*, red), as was observed in the IR experiments.

## Phosphorylation and Tpx2 have synergistic effects on AurA conformation and nucleotide binding

We used steady-state fluorescence to measure the equilibrium dissociation constants of ADP and Tpx2 for unphosphorylated and phosphorylated AurA (*Figure 2c*). Importantly, ADP bound to phosphorylated and unphosphorylated AurA with similar affinities (*Figure 2c*, top left panel), indicating that the interaction of the kinase with nucleotide is not substantially affected by phosphorylation on the activation loop. This is consistent with our IR and TR-FRET experiments, which show that phosphorylation by itself fails to trigger the long-range conformational change presumably required to couple the phosphorylation site on the activation loop to the distant ATP-binding site. In contrast, Tpx2, which does trigger a conformational change from the DFG-Out to the DFG-In state in both unphosphorylated and phosphorylated AurA, also enhances the binding affinity of nucleotides in both cases (*Figure 2c*, compare top panels).

While phosphorylation does not affect nucleotide binding to apo AurA, we found that it does substantially enhance the binding of Tpx2 to the exterior surface of the kinase, increasing the affinity by a factor of ~20 (*Figure 2c*, bottom panels). This is remarkable considering that phosphorylation does not appear to stabilize the DFG-In state, and that there are no direct contacts between the T288 residue and Tpx2 (*Bayliss et al., 2003*). In addition, once Tpx2 is bound, phosphorylation *does* lead to an enhancement of nucleotide affinity (*Figure 2c*, top right panel, compare red and blue), indicating that allosteric coupling between the phosphorylation site and the active site, missing in apo AurA, is established in the AurA:Tpx2 complex. These trends in the affinity data are in good agreement with previous enzyme kinetics measurements (*Dodson and Bayliss, 2012*). Interestingly, the synergy observed between Tpx2 and phosphorylation is also reflected in our TR-FRET experiments (*Figure 2b*). A comparison between the unphosphorylated and phosphorylated samples bound to Tpx2 shows that while the unphosphorylated sample requires nucleotide to fully shift to the active state, Tpx2 alone is sufficient to achieve this in phosphorylated AurA, and the further addition of nucleotide has little effect (*Figure 2b*, compare yellow and blue). The same trend was observed in steady-state FRET experiments (*Figure 2—figure supplement 2c*, double-headed arrows). Together these data suggest a model in which the allosteric effects of phosphorylation are somehow masked in apo AurA, and only become apparent when Tpx2 switches the kinase to the DFG-In state, at which point phosphorylation further stabilizes this state.

## Phosphorylation promotes a single functional conformation in the DFG-In state

While our results reveal synergy between phosphorylation and Tpx2, they do not answer the key question of how phosphorylation itself activates AurA. Indeed, the IR and FRET data clearly show that phosphorylation on T288 by itself does not cause a substantial shift towards the DFG-In state, and that the phosphorylated kinase, like the unphosphorylated enzyme, instead samples a range of different conformations spanning the DFG-In and DFG-Out states. We hypothesized that phosphorylation must instead drive catalytic activation of AurA by altering the structure and dynamics of the DFG-In subpopulation, presumably allowing it to populate catalytically competent geometries.

To provide insight into how phosphorylation alters the structure and dynamics of the DFG-In state, we performed molecular dynamics simulations of the wild-type kinase. Simulations were initiated from the X-ray structure of DFG-In AurA bound to ADP and Tpx2 (PDB ID: 1OL5) (*Bayliss et al., 2003*), and were run in the presence and absence of Tpx2 and with and without phosphorylation on T288. For each of these four biochemical states, 250 trajectories up to 500 nanoseconds in length were obtained on the distributed computing platform Folding@home, for a total of over 100 microseconds of aggregate simulation time for each biochemical state. Analysis of the DFG conformation revealed that the simulations remained predominantly in their initial DFG-In state (*Figure 3—figure supplement 1*), suggesting that the simulation time was insufficient to capture the slow conformational change to the DFG-Out state. The simulations can thus be regarded as probing the conformational dynamics of the DFG-In kinase.

The T288 phosphorylation site lies in the C-terminal segment of the activation loop, the correct positioning of which is essential for the binding of peptide substrates (*Figure 3a*). In the crystal structure used to initiate the simulations, this segment of the loop appears to be stabilized by interactions between the pT288-phosphate moiety and three arginine residues: R180 from the αC helix,

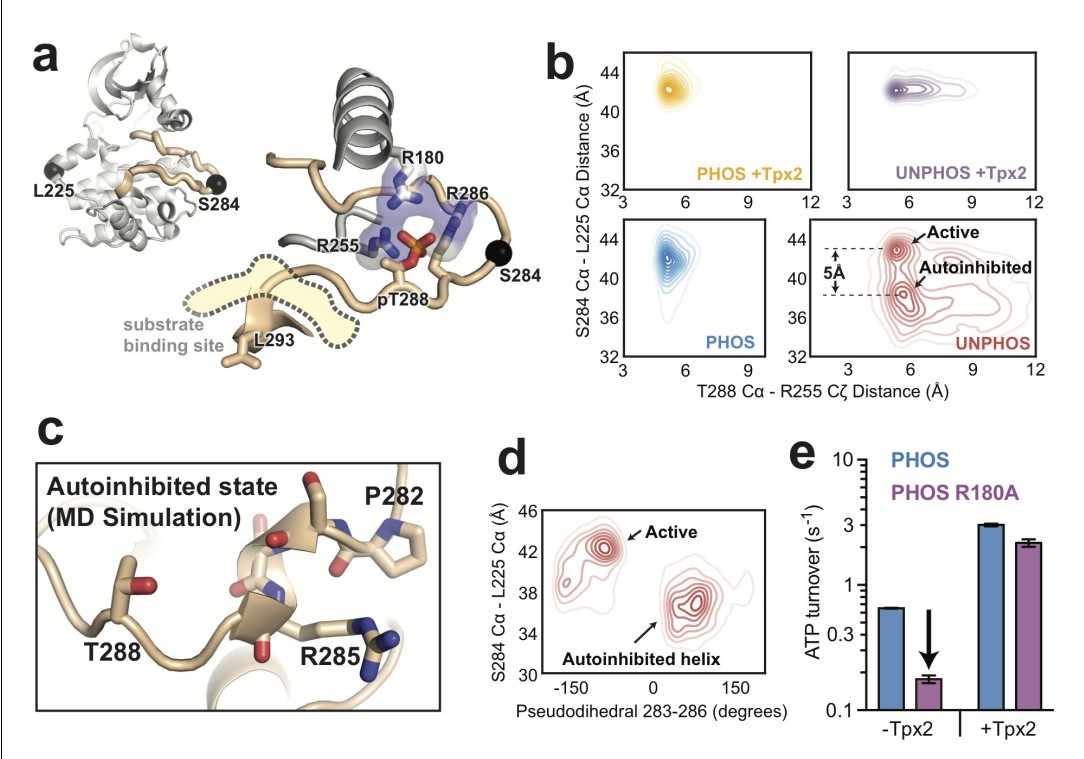

**Figure 3.** Molecular dynamics simulations of AurA show that phosphorylation disfavors an autoinhibited DFG-In substate and promotes a fully-activated configuration of the activation loop. (**a**) Structure of active, phosphorylated AurA bound to Tpx2 and ADP (PDB ID: 1OL5) showing the interactions between pT288 and the surrounding arginine residues. The S284 and L225 Cα atoms are shown as black spheres. (**b**) Contour plots showing the L225 Cα - S284 Cα distances plotted against the T288 Cα - R255 Cζ distances for all four biochemical conditions. The active and autoinhibited DFG-In states observed for the unphosphorylated kinase in the absence of Tpx2 (red), and the shift in the L225-S284 distance between them, are indicated. (**c**) Simulation snapshot showing the helical turn in the activation loop and the position of the T288 sidechain at the C-terminal end of the helix. (**d**) The L225 - S284 distance is plotted against the dihedral angle defined by the Cα atoms of residues 283–286 (pseudodihedral). The helical conformation in the autoinhibited state is indicated. (**e**) Kinase activity (shown as ATP turnover per second) for phosphorylated WT (blue) and phosphorylated R180A (purple) AurA unlabeled FRET constructs in the presence and absence of 10 µM Tpx2. The decrease in the activity in the absence of Tpx2 due to the R180A mutation is highlighted by the arrow. Data represent mean values ± s.d.; n = 3.

DOI: https://doi.org/10.7554/eLife.32766.009

The following figure supplements are available for figure 3:

**Figure supplement 1.** Analysis of DFG motif conformation and Arginine-pT288 interactions in MD trajectories.
DOI: https://doi.org/10.7554/eLife.32766.010

**Figure supplement 2.** Crystal lattice contacts in the common hexagonal crystal form of AurA (left) result in a distorted conformation of the activation loop (right), in which the peptide binding site, highlighted as sticks, is not properly assembled (structure shown in blue).
DOI: https://doi.org/10.7554/eLife.32766.011

R286 from the activation loop, and the highly conserved R255 from the catalytic loop 'HRD motif' (*Figure 3a*) (*Bayliss et al., 2003*). To probe the integrity of these interactions in the simulations, and to investigate loop dynamics in their absence, we examined the distribution of distances between the Cζ atoms of either R180 or R255 and the Cα atoms of T288 following equilibration within the DFG-In state (*Figure 3—figure supplement 1b*). We also tracked the distance between the L225 and S284 Cα atoms (the sites used for incorporating spectroscopic probes) to capture movements of the activation loop along a roughly orthogonal axis across the active site cleft.

We plotted the simulated L225-S284 distances as a function of the R255-T288 distance to assess how Tpx2 and phosphorylation affect the conformation of the activation loop (*Figure 3b*). As expected, the simulations of unphosphorylated AurA without Tpx2 show that the activation loop is highly dynamic, reflected as relatively broad distributions of L225-S284 and R255-T288 distances (*Figure 3b*, bottom right panel). The N-terminal lobe of the kinase was particularly dynamic in these simulations, and local unfolding occurred within the αC-helix in many of the trajectories, as seen

previously in simulations of the epidermal growth factor receptor (*Shan et al., 2012*) as well as in X-ray structures of the related AGC-family kinase Akt in the unphosphorylated state (*Yang et al., 2002a*). In striking contrast, the simulations showed that phosphorylated AurA is locked into a single conformation with a long L225-S284 distance (42 Å, cf. 41 Å in the 1OL5 x-ray structure) and short R255-T288 distance (~5 Å, cf. 5.5 Å in 1OL5), indicative of a stable active state in which the loop is fully ordered and the phosphothreonine residue forms ion-pairing interactions with R255 and R180 (*Figure 3b*, bottom left panel, and *Figure 3—figure supplement 1b*). Interestingly, phosphorylation alone is almost as effective at constraining the loop in the active state as phosphorylation and Tpx2 together (*Figure 3b*, left panels). In contrast, the simulations show that the activation loop of unphosphorylated AurA bound to Tpx2 remains somewhat dynamic (*Figure 3b*, top right panel), and additional phosphorylation significantly stabilizes the loop.

## Phosphorylation disrupts an autoinhibitory DFG-In substate

Although unphosphorylated AurA is highly dynamic in the absence of Tpx2, the activation loop is not in fact disordered in the simulations. Instead, two discrete subpopulations are observed: one subpopulation corresponding to the active-like state with a long L225-S284 distance (~43 Å), and another with a much shorter distance (~38 Å), representing a DFG-In conformation in which the activation loop is not correctly positioned for catalytic function (*Figure 3b*, lower right panel). Manual inspection of the trajectories revealed that in this subpopulation the tip of the activation loop folds into a short helical turn spanning residues P282-R286, with the P282 proline residue serving as the N-terminal capping residue (*Kumar and Bansal, 1998*) in most of the trajectories (*Figure 3c*). Calculating the pseudodihedral angle for the Cα atoms of S283-R286 across all trajectories confirmed that the inactive subpopulation possessed well-defined helical pseudodihedral values of 50–75° (*Figure 3d*). Although this conformation has not been observed in X-ray structures of AurA, the formation of short helices in the activation loop is a common feature of the inactive states of other protein kinases (*Sicheri et al., 1997*; *Wood et al., 2004*; *Lee et al., 2010*; *De Bondt et al., 1993*).

An interesting feature of the autoinhibited DFG-In substate observed in the simulations is that the T288 residue, which immediately follows the helical segment in the protein sequence, is positioned close to the C-terminal end of the helix in almost all of the trajectories (*Figure 3c*), with the sidechain hydroxyl forming hydrogen bonds to the backbone carbonyls of residues R285 and R286 in many of the simulation snapshots. We reasoned that upon phosphorylation of T288, the proximity of the phosphate group to the negatively-charged end of the helix dipole (*Hol et al., 1978*) would destabilize this autoinhibited substate, promoting the refolding of the activation loop to the active conformation.

We wondered why the helical conformation of the activation loop has not been observed in X-ray structures of AurA. In fact, the activation loop adopts the active conformation in only a small subset of AurA structures, specifically those determined either in the presence of Tpx2 (*Bayliss et al., 2003*; *Zhao et al., 2008*; *Clark et al., 2009*) or other protein factors that stabilize the active state (*Richards et al., 2016*). Instead, almost all of the structures of AurA in the DFG-In state (76 structures out of 138 total structures of AurA in the PDB) were determined in the same hexagonal crystal form in which the kinase adopts an inactive conformation with the activation loop misaligned and the peptide binding site disassembled. Upon examination of the crystal lattice we noticed that this conformational state of the activation loop appears to be induced by a crystal contact between the peptide binding site and a neighboring molecule in the lattice (*Figure 3—figure supplement 2*). This apparent crystallographic artifact may have prevented previous observation of the helical autoinhibited DFG-In substate visualized in our simulations, which model the kinase in solution rather than in the crystallographic context.

Our MD simulations, which represent over a millisecond of simulation data, predict that phosphorylation has profound effects on the activation-loop conformation of AurA within the DFG-In state, both disrupting an autoinhibited substate and promoting an active substate that is primed for catalytic function. In an attempt to confirm the simulation result that the T288 phosphothreonine residue of phosphorylated AurA is correctly coordinated in the active state, we mutated its ion-pairing partner R180 to an alanine residue and measured the effect on kinase activity. The R180A mutant possessed 4-fold lower activity in the absence of Tpx2, whereas the activity in the presence of Tpx2 was only modestly affected (*Figure 3e*). This is consistent with the catalytic activity of

phosphorylated AurA arising from a population of molecules adopting the canonical active state in which the phosphothreonine residue is correctly ion-paired.

## DEER experiments confirm that phosphorylation alters the structure of the DFG-In state, rather than shifting the DFG equilibrium

To assess the effects of phosphorylation on the structure of the DFG-In state predicted by our simulations, we used double electron-electron resonance (DEER), a pulsed electron paramagnetic resonance (EPR) spectroscopy technique (*Jeschke, 2012*). DEER experiments measure the dipole-dipole interactions of unpaired electron spins to provide high-resolution information about the distribution of spin-spin distances in the sample. Two MTSL spin labels were incorporated into AurA at the same positions used for FRET experiments (L225C and S284C). Labeling and phosphorylation were confirmed by mass spectrometry, and MTSL-labeled samples retained close to full kinase activity (*Figure 4—figure supplement 1*). Samples were flash frozen in the presence of saturating concentrations of nucleotides and/or Tpx2, and DEER experiments were performed at 65 K.

We first sought to confirm our above results that the phosphorylated kinase still samples both the DFG-Out and DFG-In states. DEER spectra (background-corrected dipolar evolution data) take the form of a damped oscillating signal in which the mean distance between the spin labels is encoded in the cube root of the oscillation period, and the width of the distance distribution is encoded in the degree of damping. DEER spectra acquired for the phosphorylated kinase bound to ADP or AMPPNP showed a rapidly decaying and heavily damped signal, consistent with the activation loop adopting multiple conformations (*Figure 4a* and *Figure 4—figure supplement 2a*, blue lines). Extraction of spin-spin distances from the DEER spectra using Tikhonov regularization (*Chiang et al., 2005*) confirmed the presence of a broad distribution of distances, spanning ~35 to~55 angstroms (*Figure 4b*, blue). Control experiments with the model peptide substrate kemptide, which conforms to the consensus phosphorylation site sequence for AurA (*Ferrari et al., 2005*), showed no effect on the conformational ensemble of the phosphorylated kinase (*Figure 4—figure supplement 2b*). However, DEER spectra of phosphorylated AurA bound to Tpx2 decayed much less rapidly, indicating increased spin-spin distance, and exhibited pronounced oscillations, indicating a high degree of structural order (*Figure 4a*, yellow). The corresponding distance distribution displayed a single dominant peak at 52 angstroms (*Figure 4b*, yellow), representing a ~5 angstrom longer mean spin-spin distance than seen in the samples lacking Tpx2, consistent with the activation loop now adopting the extended DFG-In conformation (see *Figure 2a*).

To bolster these DEER experiments we performed molecular dynamics simulations of MTSL-labeled phosphorylated AurA in either the DFG-Out state (PDB ID: 5L8K) or the fully-active DFG-In state (PDB ID: 1OL5, AurA bound to Tpx2), totaling 75–110 microseconds of aggregate simulation data for each state. In these simulations, sampling of different spin label rotamers, combined with motion of the protein, gives rise to a range of predicted spin-spin distances. Although the simulated distributions for the DFG-In and DFG-Out states overlap, the DFG-In distribution is skewed to longer distances: distances beyond 45 angstroms are more populated in the DFG-In simulations, and distances beyond 50 angstroms are almost exclusively associated with the DFG-In state (*Figure 4c*). Prominent peaks are present in the DFG-In distribution at ~44,~48 and~52 angstroms (*Figure 4c*). The 44- and 52-angstrom predicted spin-spin distances are also observed in the DEER experiment performed with phosphorylated AurA bound to Tpx2 (*Figure 4b*, yellow), whereas the 48-angstrom peak was not observed experimentally; presumably the corresponding rotamer state is too sparsely populated to be detected at the low temperature of the DEER experiment. The simulations nonetheless confirm that the 52-angstrom spin-spin distance, which is so prominent in the experiment performed in the presence of Tpx2, almost certainly arises from the DFG-In state, and underscore that in the absence of Tpx2 phosphorylated AurA occupies a mixture of DFG-In and DFG-Out states, consistent with the IR and FRET experiments above.

We next turned to assessing how phosphorylation alters the structure of the DFG-In state. DEER experiments performed on unphosphorylated AurA bound to ADP gave qualitatively similar results to those obtained with the phosphorylated kinase bound to ADP, with a broad distribution of spin-spin distances consistent with similar populations of DFG-In and DFG-Out states (*Figure 4b*, compare red and blue). However, close inspection of the DEER spectra revealed subtle differences between the unphosphorylated and phosphorylated samples (*Figure 4a* inset), and the corresponding distributions indicated that distances beyond 50 angstroms were more populated in the

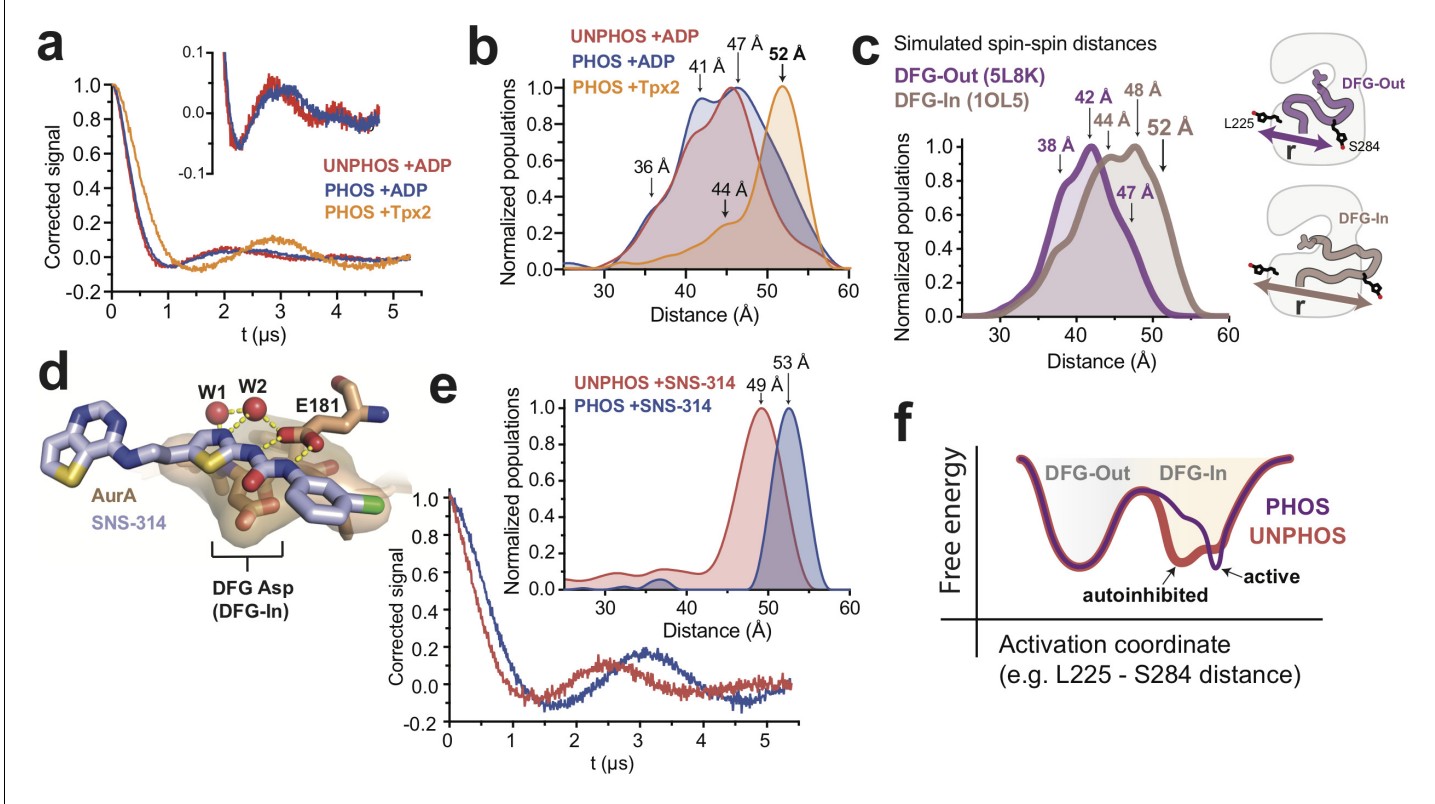

**Figure 4.** DEER spectroscopy confirms that phosphorylation of AurA alters the DFG-In state. (a) Background-corrected DEER spectra of unphosphorylated AurA bound to ADP (red), and phosphorylated AurA bound to either ADP (blue) or to Tpx2 (yellow). The inset shows an enlarged view of the spectra for the +ADP samples. (b) Population densities obtained by Tikhonov regularization for the data shown in (a), with prominent peaks in the distributions indicated. The increased sampling of distances beyond ~50 angstroms in the phosphorylated kinase bound to ADP is highlighted with darker blue shading. Data are from single representative experiments of two independent repeats. (c) Spin-spin distance distributions obtained by molecular dynamics simulations initiated from X-ray structures of AurA in either the DFG-Out state (purple) or the fully-active DFG-In state with both Tpx2 and phosphorylation (pink). The inset shows schematics of the spin-labeling scheme. (d) X-ray structure of SNS-314 bound to AurA highlighting interactions with the DFG motif, structured water molecules and the catalytic glutamate (E181) specific to the DFG-In state (PDB ID: 3D15). (e) DEER spectra (main panel) and distance distributions (inset) measured for unphosphorylated (red) and phosphorylated (blue) AurA bound to SNS-314. The distributions are vertically aligned with those shown in (b) to facilitate comparison. (f) Hypothesized energy landscape for AurA, highlighting the effect of phosphorylation on the DFG-In state.
DOI: https://doi.org/10.7554/eLife.32766.012

The following figure supplements are available for figure 4:

**Figure supplement 1.** Validation of spin-labeled AurA constructs used for EPR.
DOI: https://doi.org/10.7554/eLife.32766.013

**Figure supplement 2.** DEER experiments with AMPPNP and peptide substrate.
DOI: https://doi.org/10.7554/eLife.32766.014

**Figure supplement 3.** DEER experiments with an alternate spin labeling site support a phosphorylation-driven structural transition in the DFG-In state.
DOI: https://doi.org/10.7554/eLife.32766.015

phosphorylated sample (*Figure 4b*, dark blue shading). The same trend was observed in experiments performed with AMPPNP instead of ADP (*Figure 4—figure supplement 2a*). We wondered whether the presence of a substantial DFG-Out subpopulation in these samples might be obscuring a more dramatic structural change occurring in the DFG-In subpopulation. To test this, we used the ATP-competitive AurA inhibitor SNS-314, which preferentially binds to the DFG-In state of AurA (*Oslob et al., 2008*) (*Figure 4d*), to induce a homogeneous population of DFG-In kinase. Strikingly, DEER spectra measured on unphosphorylated and phosphorylated AurA bound to SNS-314 showed pronounced oscillations with differing periods, indicating that both samples adopt well-defined structures, but that the major spin-spin distances are different in the two cases (*Figure 4e*). Indeed,

the Tikhonov distributions showed sharp peaks at 49 and 53 angstroms for the unphosphorylated and phosphorylated kinase, respectively (*Figure 4e*, inset).

Contrasting these results with those obtained with ADP (*Figure 4b*) highlights that the binding of SNS-314 stringently enforces adoption of the DFG-In state and dramatically simplifies the conformational ensemble. Importantly, the resulting unobstructed view of the DFG-In state shows that phosphorylation does indeed lead to a pronounced structural change of the activation loop, reflected in the increase in spin-spin distance from 49 to 53 angstroms shown in *Figure 4e*. The distance distribution obtained for phosphorylated AurA bound to SNS-314 (*Figure 4e*, blue) is very similar to that observed in the presence of Tpx2 (*Figure 4b*, yellow), confirming that these longer spin-spin distances (52–53 Å) arise from the catalytically active DFG-In conformation, whereas the shorter spin-spin distance observed with unphosphorylated AurA bound to SNS-314 (~49 Å) corresponds to a structurally-distinct DFG-In conformation. This phosphorylation-driven structural change was also detected in a separate set of DEER experiments in which the S284 labeling site was moved to S283, and SNS-314 again used to isolate the DFG-In state (*Figure 4—figure supplement 3*).

Interestingly, the 4-angstrom increase in spin-spin distance observed in the DEER experiments performed with the L225/S284 labeling sites (*Figure 4e* inset) is similar to the difference in the L225-S284 Cα distances predicted by the MD simulations for the autoinhibited and active DFG-In substates (*Figure 3b*, bottom right panel). The DEER experiments thus support the model that phosphorylation triggers a switch from an autoinhibited DFG-In substate to the active DFG-In substate (see *Figure 3c*). We conclude that while phosphorylated AurA samples the DFG-Out and DFG-In states to a similar extent as the unphosphorylated kinase, the structure and dynamics of the DFG-In subpopulation are profoundly altered by phosphorylation, leading to catalytic activation.

Our proposed model for the autoinhibited DFG-In substate of AurA provides an explanation for several puzzling observations. Firstly, the presence of a third state explains how phosphorylation can promote the active state without triggering a shift in the DFG equilibrium, the central result of this paper. By eliminating the autoinhibited DFG-In substate, phosphorylation redistributes the ensemble between the DFG-Out and active DFG-In substates (*Figure 4f*). The fact that this does not substantially change the DFG equilibrium suggests that the stabilizing ionic interactions between the phosphothreonine and the arginine residues (see *Figure 3a*) are offset by an energetic penalty associated with refolding the activation loop into the active configuration.

Secondly, our model accounts for the synergy between phosphorylation and Tpx2, observed in our fluorescence experiments, and reported previously (*Dodson and Bayliss, 2012*; *McIntyre et al., 2017*). By unfolding the activation loop from the autoinhibited DFG-In state, phosphorylation promotes the formation of the Tpx2-interaction surface by the N-terminal segment of the activation loop, resulting in dramatically enhanced binding affinity. By binding tightly to this surface, Tpx2 in turn compensates for the energetic cost of reconfiguring the activation loop, allowing the effects of phosphorylation to be manifested as a further stabilization of the active DFG-In state, as we observed in our fluorescence experiments in terms of an additional conformational shift and enhanced nucleotide binding. While definitive confirmation of the structural model for the autoinhibited DFG-In state awaits further experiments, our DEER data are fully consistent with this model, which provides a mechanism for the activation of AurA by phosphorylation that accounts for the available structural, dynamic and biochemical data.

## Discussion

The majority of eukaryotic protein kinases are activated by phosphorylation on the activation loop at a site equivalent to T288 in AurA (*Johnson et al., 1996*). X-ray structures have suggested that the functional role of this phosphorylation is to trap the kinase in an active DFG-In state and rigidify the flexible activation loop in a specific configuration that promotes catalysis and substrate binding (*Knighton et al., 1991*; *Yamaguchi and Hendrickson, 1996*; *Steichen et al., 2012*). Our results show that phosphorylation can drive catalytic activation of a protein kinase without restraining the protein in the DFG-In state, providing a contrasting and highly dynamic view of an activated kinase in which major conformational changes of catalytic elements may occur continuously during the catalytic cycle. Of note, a recent single-molecule fluorescence study also reported that phosphorylated AurA dynamically transitions between multiple structural states (*Gilburt et al., 2017*).

Binding of Tpx2 to unphosphorylated AurA causes a pronounced population shift from the DFG-Out to the DFG-In state (*Cyphers et al., 2017*), in striking contrast with the phosphorylation-mediated activation mechanism described here. Our simulation data also reveal differences in how phosphorylation and Tpx2 affect the DFG-In subpopulation, with Tpx2 less effective at constraining the C-terminal segment of the activation loop. Thus it appears that phosphorylation and Tpx2 activate AurA through quite different - albeit complementary - mechanisms, with phosphorylation triggering a structural switch within the DFG-In subpopulation, and Tpx2 instead promoting a DFG flip to increase the population of the DFG-In state. Although we have not explicitly tracked the dynamics of the important αC-helix in this work, our previous work showed that the binding of Tpx2 substantially stabilized the αC-helix and the associated regulatory spine (*Cyphers et al., 2017*). Given that the R180 residue on the αC-helix directly recognizes the pT288 phosphothreonine residue, and that the R180A mutation interferes with activity of the phosphorylated enzyme, phosphorylation is presumably also coupled to the αC-helix. However, while this coupling likely contributes to the switch from the autoinhibited DFG-In substate to the active DFG-In substate, our spectroscopic experiments show, surprisingly, that these interactions are not sufficiently stabilizing to substantially increase the overall population of the DFG-In state. Thus, phosphorylation can be thought of as tuning the free energy surface for the DFG-In state - favoring the active substate over the autoinhibited substate - as opposed to changing the relative free energies of the DFG-In and DFG-Out states.

It is interesting to consider the dual activation mechanisms of AurA in the light of the closely-related AGC-family kinases. The AGC kinases possess a C-terminal hydrophobic motif that docks in cis onto a pocket above the αC-helix in a manner that closely resembles the interaction of Tpx2 with AurA (*Frödin et al., 2002*; *Yang et al., 2002b*). Activation of the AGC kinases by activation loop phosphorylation and hydrophobic motif engagement are tightly coupled events (*Alessi et al., 1996*) and are both important for activation (*Batkin et al., 2000*). The contrasting ability of AurA to be independently activated by phosphorylation or Tpx2 is dependent upon a unique active-site water network that strengthens the regulatory spine of the kinase relative to that of the AGC kinases (*Cyphers et al., 2017*). Interestingly, when Tpx2 and phosphorylation act together, they are capable of overriding the deleterious effects of mutations in the regulatory spine designed to disrupt the water network and render the spine more like that of an AGC kinase (*Cyphers et al., 2017*). Presumably the synergistic effects of Tpx2 and phosphorylation on the conformational dynamics of the kinase, in which the αC-helix is stabilized by Tpx2 engagement and the activation loop is locked into the active DFG-In conformation, renders the additional stabilization provided by the water network redundant. It is likely that the fully-activated AGC kinases, with their hydrophobic motifs engaged on the αC-helix and their activation loops phosphorylated, closely resemble this conformationally-rigid form of AurA.

In contrast, the highly dynamic nature of AurA activated only by phosphorylation may be a unique property of the Aurora kinases that reflects the loss of the hydrophobic motif in this lineage. These motions may facilitate further regulation of AurA by additional cellular factors, allowing for graded levels of catalytic activity. For instance, phosphorylated AurA has been shown to interact with Cep-192 (*Joukov et al., 2014*), Bora (*Macůrek et al., 2008*; *Seki et al., 2008*), and Ajuba (*Hirota et al., 2003*) at the centrosome, and these interactions can further regulate AurA activity towards specific substrates. Although a model peptide substrate did not modulate AurA dynamics in our experiments, in the context of the higher-order signaling complexes found at the centrosome substrates may be presented at sufficiently high local concentration to further stabilize the activation loop in the active conformation and promote phosphoryl transfer. In this context it is noteworthy that the oncogenic transcription factor N-Myc binds to AurA through a pseudosubstrate interaction, and does indeed appear to stabilize the phosphorylated kinase in the active conformation (*Richards et al., 2016*). These observations are consistent with the overarching view that protein kinases evolve under strong selective pressure to optimize their responsiveness to regulatory inputs, and not for maximal catalytic efficiency.

It is unclear whether activation of AurA by both phosphorylation and Tpx2 occurs in normal cells, where the Tpx2-bound spindle pool of AurA is thought to be predominantly unphosphorylated (*Zeng et al., 2010*; *Toya et al., 2011*), and the centrosomal pool is not bound to Tpx2 (*Kufer et al., 2002*). The doubly-activated form of the kinase is prominent, however, in the tumors of ~10% of melanoma patients, where mutational inactivation of the PP6 phosphatase leads to accumulation of phosphorylated AurA bound to Tpx2 on the mitotic spindle, resulting in chromosome instability and

DNA damage that can be partially reversed by AurA inhibitors (*Hammond et al., 2013*; *Hodis et al., 2012*; *Gold et al., 2014*). The distinct conformational dynamics of doubly-activated AurA might provide opportunities for the development of improved inhibitors that selectively target this form of the kinase in melanoma cells.

The DFG flip has long been considered one of the key regulatory mechanisms used by nature to control the catalytic activity of protein kinases (*Hubbard et al., 1994*; *Nagar et al., 2003*), but the difficulty of directly observing this structural change in solution and correlating it with activity has hampered efforts to conclusively demonstrate its regulatory role. Although AurA does adopt the DFG-Out state, our results show that activation of the enzyme by phosphorylation is not mediated by a DFG flip, but rather by inducing activating conformational changes within the DFG-In state. It is noteworthy that many other protein kinases, including non-receptor and receptor tyrosine kinases (*Wood et al., 2004*; *Xu et al., 1997*) and the cyclin-dependent kinases (*De Bondt et al., 1993*), employ an autoinhibitory DFG-In state, as opposed to the DFG-Out state, for regulatory control. A major remaining question is whether the substantial DFG-Out subpopulation of phosphorylated AurA performs an important biological function, and whether this is a unique feature of AurA itself, or a more general property of activated protein kinases.

## Materials and methods

### Expression and purification of AurA constructs

Aurora A kinase domain constructs were expressed in *E.coli* and purified as previously described (*Cyphers et al., 2017*). We used a Cys-lite form of AurA (C290A C393S) that has been previously shown to possess robust kinase activity within 2-fold of WT AurA, and can be purified from bacteria in homogeneously T288-phosphorylated form (*Burgess and Bayliss, 2015*; *Rowan et al., 2013*). For IR and EPR experiments it was necessary to mutate an additional cysteine (C247) to alanine to avoid non-specific labeling. Site-directed mutagenesis was performed using the QuikChange Lightning kit (Agilent). Phosphorylation was verified by mass spectrometry and activity assays.

### IR spectroscopy

Phosphorylated Q185C AurA protein samples (human AurA residues 122–403 containing an N-terminal hexahistidine tag) were prepared using a cysteine-light form of the kinase in which all endogenous solvent-accessible cysteines were removed by mutagenesis (Q185C, C247A, C290A, C393S). After Nickel-affinity purification, repeated rounds of cation exchange chromatography were used to isolate the homogeneous singly phosphorylated species, with enrichment of the phosphorylated species tracked during purification by western blotting and activity assays (*Figure 1—figure supplement 1*). Nitrile labeling of the purified protein was performed using a 1.5:1 molar ratio of DTNB (Ellman's reagent), followed by 50 mM KCN, and excess labeling reagents were removed using a fast desalting column (GE Healthcare). Incorporation of a single nitrile label was confirmed by whole-protein mass spectrometry (*Figure 1—figure supplement 1*). Samples for IR spectroscopy were prepared by concentrating labeled protein (50–100 μM) in the presence or absence of 4 mM ADP and 8 mM MgCl$_2$, and/or excess Tpx2 peptide (~150 μM, residues 1–43 of human Tpx2, Selleckchem) in FTIR buffer (20 mM Hepes, pH 7.5, 300 mM NaCl, 20% glycerol). Samples were concentrated to ~1 mM and loaded into a calcium fluoride sample cell mounted in a temperature-controlled housing (Biotools) for IR experiments. IR spectra were recorded on a Vertex 70 FTIR spectrometer (Bruker) equipped with a liquid nitrogen cooled indium antimonide detector with 2 cm$^{-1}$ spectral resolution. Spectra were averaged across 256 scans, background subtracted using absorbance spectra of the buffer flow-through from sample concentration, and baselined using the polynomial method in the OPUS software (Bruker).

### Kinase assays

Kinase activity was measured using the ADP Quest coupled kinase assay (DiscoverX) in a fluorescence plate reader (Tecan Infinite M1000 PRO) as described previously (*Cyphers et al., 2017*). Assays were performed using 2, 5, 10, 100, or 200 nM kinase (depending on the protein variant), 1 mM kemptide peptide substrate (Anaspec), 10 μM Tpx2 residues 1–43 (Selleckchem), and 50 μM ATP (Sigma Aldrich). Activity was determined by fitting the fluorescence intensity as a function of

time using linear regression. Background ATPase activity was determined using samples lacking peptide substrate, and was subtracted from the activity in the presence of substrate peptide. The average fluorescence for several ADP concentrations in the dynamic range of the assay was used to construct a calibration curve, and convert the background-corrected activity to ATP turnover numbers. Activities given are the average of three experiments, where error bars are the standard deviation of the replicates.

## Fluorescence and Förster resonance energy transfer (FRET) experiments

For FRET experiments, the AurA variant C290S/A C393S L225C S284C was expressed, purified and labeled with donor (Alexa 488, Invitrogen) and acceptor (Alexa 568) using cysteine-maleimide chemistry, as previously described (*Cyphers et al., 2017*). Labeled samples were validated using activity assays and mass spectrometry and retained close to full kinase activity (*Figure 2—figure supplement 1*).

## Steady-state FRET

Ligand titration FRET experiments were performed in a Fluoromax 4 Spectrofluorometer (Horiba) at 22°C. Assays were performed in 15 mM HEPES pH 7.5, 20 mM NaCl, 1 mM EGTA, 0.02% Tween-20, 10 mM MgCl$_2$, 1% DMSO and 0.1 mg/mL bovine-γ-globulins at AurA concentrations of 5–50 nM. Bulk FRET efficiency and inter-fluorophore distance were calculated from the ratios of the donor fluorescence in the presence and absence of acceptor, assuming a value of 62 angstroms for the Förster radius, as previously described (*Cyphers et al., 2017*). The steady-state anisotropy was below 0.2 for all samples and was similar across biochemical conditions (*Figure 2—figure supplement 2b*).

Dissociation constants $K_D$ were determined using the spectra obtained with the D + A labeled sample. The ratio $F_D/F_A$ (where $F_D$ is the donor fluorescence maximum and $F_A$ is the acceptor fluorescence maximum), a highly sensitive measure of ligand binding, was fit as a function of ligand concentration. For calculation of the $K_D$ for Tpx2 with saturating ADP bound, $K_D$ is near the concentration of fluorescent protein. Ligand depletion was accounted for by fitting the raw data to determine the plateau $F_D/F_A$ value (representing ligand saturation), calculating the percent saturation for each total ligand concentration, and then back calculating the concentration of free ligand. $F_D/F_A$ was then re-fit as a function of free ligand concentration.

## Time-resolved (TR) FRET

The time-correlated single photon counting instrument used to collect time-resolved fluorescence data has been previously described (*Agafonov et al., 2009*). The instrument response function was obtained with the emission polarization set at vertical, while fluorescence data were collected with the emission polarization set at 54.7°, with a GFP band pass filter in place (Semrock).

Experimental buffer contained 15 mM HEPES pH 7.5, 20 mM NaCl, 1 mM EGTA, 0.02% Tween-20, 10 mM MgCl$_2$. Experiments were performed at 100–200 nM unphosphorylated or phosphorylated FRET-labeled AurA, in the presence and absence of 100–125 μM Tpx2 and 0.2–1 mM ADP or AMPPNP; one phosphorylated AurA experiment also contained 1 mM DTT. For both unphosphorylated and phosphorylated AurA, three independent experiments were performed and analyzed.

Data fitting was performed as previously described (*Muretta et al., 2013*). Briefly, time-resolved fluorescence waveforms were fit using custom software designed for analysis of time-resolved fluorescence (*Muretta et al., 2010*). The instrument response function and the model of the fluorescence decay were convolved to fit the measured time-resolved fluorescence waveform. Donor-only fluorescence waveforms were modeled using a multiexponential decay function, which accounts for the intrinsic lifetimes of the dye, with two exponentials required to fit the Alexa 488 fluorescence decay. Donor +acceptor (D + A) waveforms were modeled from the amplitudes and lifetimes present in the matched donor-only sample and modified so that a distance-dependent resonance energy transfer term, corresponding to a Gaussian distribution of inter-probe distances, describes the decrease in fluorescence lifetime relative to the donor-only control. The mean distance and full-width half maximum of the Gaussian functions were fit individually for each D + A and D-only pairing, while the parameters that described general conditions of the experiment common among all samples, such as the fraction of a given D + A sample lacking acceptor dye, were globally linked.

## EPR experiments

DEER samples were prepared using the Cys-lite mutant construct of AurA C290S/A C393S C247A L225C S284C (or S283C), purified as described above. AurA was labeled with MTSL (Santa Cruz Biotechnology), purified by cation exchange chromatography, and concentrated. Labeling was verified using mass spectrometry, and MTSL-labeled samples retained close to full activity of unlabeled AurA in the presence and absence of Tpx2 (*Figure 4—figure supplement 1*). The protein was then buffer exchanged into the experimental buffer, which was 20 mM HEPES pH 7.5, 300 mM NaCl, 10% deuterated glycerol, 2% v/v $H_2O$ in $D_2O$. For DEER experiments, samples containing 30–60 μM MTSL-labeled AurA were prepared in the presence and absence of 100–200 μM Tpx2 and 300 μM ADP (8 mM $MgCl_2$ was added to samples containing ADP). Final samples varied in v/v $H_2O$ concentration from 2–14%; however, no significant differences were observed in Tikhonov distributions derived from experiments performed in protonated and deuterated buffers. Samples were flash-frozen in an isopropanol dry ice bath followed by liquid nitrogen. Data shown are from one of two replicate experiments.

DEER spectra were detected at 65 K using an Elexsys E580 Q-Band spectrometer (Bruker Biospin) equipped with an ER 5107D2 resonator (Bruker Biospin) using the standard 4-pulse pulse sequence with π/2 and π pulses (including ELDOR) set to 16 and 32 ns respectively. The pump frequency was set to the central resonance position of the nitroxide echo-detected field swept spectrum while the observe position was set 24 G up-field to avoid excitation bandwidth overlap (*Agafonov et al., 2009*).

Data were analyzed using custom software written in Mathematica which was based heavily on DeerAnalysis 2017 (*Jeschke et al., 2006*). The raw spectra were phase and background corrected assuming a homogeneous background model to produce the DEER waveform. Distance distributions were determined using Tikhonov regularization, with an optimal smoothing parameter chosen using a combination of the l-curve and leave one out cross validation (LOOCV) techniques (*Edwards and Stoll, 2016*). After choice of smoothing parameter, a range of background fits were performed to identify stable populations in the distance distributions, with highly unstable, long distance populations being largely attributable to errors in the background fit and model choice (*Jeschke, 2011*).

## Molecular dynamics simulations

### System preparation

#### Modeling WT unphosphorylated AurA

WT AurA in complex with ADP was simulated with and without Tpx2. All simulations were started from the X-ray structure of WT AurA bound to Tpx2 and ADP in the presence of three magnesium ions (PDB ID: 1OL5 [*Bayliss et al., 2003*]). From the crystal structure, PDBFixer (https://github.com/pandegroup/pdbfixer) version 1.2 was used to model in Tpx2 residues 23–29 (unresolved in 1OL5), add hydrogens belonging to standard dominant protein residue protonation states at pH 7.4, and remove phosphorylation from threonine residues 287 and 288 (*Eastman et al., 2013*). Crystallographic waters were retained to prevent nonphysical collapse of hydrophilic pockets during minimization. The chain containing Tpx2 was then removed for simulations without Tpx2 (-Tpx2) and retained for simulations with Tpx2 (+Tpx2). Sulfate ions present in the crystal structure were manually removed. The crystallographic ADP (containing only heavy atoms) was extracted from the structure and converted to a protonated Tripos mol2 file using OpenEye toolkit OEChem v2015.June (*Marcou and Rognan, 2007*; *Stahl and Mauser, 2005*). The protein structure was then loaded as an OpenMM version 7.0.1 Modeller object, and the protonated ADP was reintroduced through conversion from mol2 to OpenMM format via MDTraj 1.4.2 (*Eastman et al., 2013*; *McGibbon et al., 2015*).

#### Modeling WT phosphorylated AurA

Simulations of phosphorylated WT AurA in complex with ADP were prepared as above, but the phosphothreonines (denoted TPO in the PDB file) at positions 287 and 288 were left in place and parameterized using the Sticht T1P AMBER parameters (*Homeyer et al., 2006*) retrieved from the AMBER parameter database (*Bryce, 2015*). The AMBER phosphothreonine parameter file was converted to OpenMM ffxml using a python script that has been made publically available (*Chodera, 2017*; copy archived at https://github.com/choderalab/AurA-materials), and subsequently

converted into a hydrogen specification file for OpenMM's Modeller by hand. The PDB file generated by PDBFixer was loaded into an OpenMM Modeller object where the hydrogens and bonds were added to the TPO residues using a Forcefield object instantiated with the AMBER99Bildn parameters as well as the custom TPO parameters described above. All crystallographic waters, ADP, sulfate ions and magnesium ions were handled as with the unphosphorylated WT AurA simulations.

## Parameterization the WT AMBER simulations

An OpenMM ForceField was instantiated using AMBER99SBildn force field parameters (*Meagher et al., 2003*) for the protein and TIP3P water model, along with ADP parameters generated by Carlson and accessed from the Amber Parameter Database (*Homeyer et al., 2006*; *Bryce, 2015*; *Meagher et al., 2003*). The phosphorylated simulations also used custom phosphothreonine parameters described above (*Homeyer et al., 2006*; *Bryce, 2015*).

## Minimization and equilibration for the WT AMBER simulations

Local energy minimization was performed in three separate steps in order to gradually introduce bond constraints. An OpenMM System was instantiated with no constraints on bonds or angles for the first minimization, which took place in vacuum (with crystallographic waters) with no constraints on bonds or angles. After this minimization, a new System was instantiated with constraints on the lengths of all bonds involving a hydrogen atom, and minimization was repeated. The structure and positions of all atoms were then put into a new OpenMM Modeller object, where TIP3P waters were added to a cubic box extending 11 Å beyond the outermost protein atoms, along with neutralizing counterions and sufficient excess NaCl to achieve an effective salt concentration of 300 mM. Another System with constrained bonds to hydrogen was created from the solvated structure and minimized. To minimally relax the structure before deploying simulations to Folding@Home, 5000 steps of Langevin dynamics were run using a Langevin integrator with a time step of 2.0 fs, temperature of 300.0 K, and collision rate of 5.0 ps$^{-1}$. Nonbonded forces were modeled using the particle-mesh Ewald (PME) method with default parameters with a cutoff distance of 9.0 Å. All other settings remained at default values, except double precision was used throughout the minimization-and-equilibration process.

## Production simulation for WT amber simulations

The resulting system, integrator, and state data from minimal equilibrations were serialized to XML format for simulation on Folding@Home using a simulation core based on OpenMM 6.3 (*Eastman et al., 2013*; *Shirts and Pande, 2000*) for both the phosphorylated and unphosphorylated systems. This entire process was repeated five times each phosphorylation simulations with and without Tpx2 to set up individual Folding@home RUNs, with each RUN representing a distinct initial configuration generated by the minimization-and-equilibration procedure. For each of the RUNs, 50 CLONEs with different initial random velocities and random seeds were simulated on Folding@home, where each clone ran for a maximum of 500 ns (250 million Langevin dynamics steps of 2 fs timestep with all-atom output frames saved every 125,000 steps using single precision and a Monte Carlo Barostat with pressure of 1 atm, temperature of 300 Kelvin, and barostat update interval of 50 steps), generating over 100 µs of aggregate simulation data for each of the WT conditions (with and without Tpx2).

## Modeling Spin Probe-labeled AurA

Because AMBER parameters for spin probes were not widely available, we used the equivalently modern CHARMM36 generation of forcefields for MTSL-labeled AurA, simulated in complex with ADP and with and without both TPX2 and phosphorylation, for a total of four possible combinations per starting structure. Simulations were started from two different starting configurations: DFG-in (PDB ID: 1OL5) and DFG-out (5L8K (*Burgess et al., 2016*)). For the crystal structure of 1OL5, Schrödinger's PrepWizard (*Sastry et al., 2013*) (release 2016–4) was used to model in Tpx2 residues 23–29 (unresolved in 1OL5), and add in hydrogens at pH 7.4 for both protein residues and ADP. The protonation state of ADP was assigned the lowest energy state using Epik at pH 7.4 ± 2. Hydrogen bonding was optimized using PROPKA at pH 7.4 ± 2. The entire structure was minimized using OPLS3 and an RMSD convergence cutoff of 0.3 Å. Because TPX2 is not present in the 5L8K

structure, the coordinates and crystal waters of Tpx2 in 1OL5 after preparation were transferred to the unprepared 5L8K after aligning the kinase domain to 1OL5 and deleting the vNAR domain (chain B) from 5L8K. 5L8K was then prepared using the same protocol as above, removing any organic solvent molecules but retaining all crystal waters. All three structures were run through CHARMM-GUI Solvator tool (*Jo et al., 2008*) (http://www.charmm-gui.org/?doc=input/solvator). In the first stage of this tool, all crystal waters and magnesiums in the structures where retained, while sulfates were deleted. Tpx2 was deleted at this stage for the non-Tpx2 conditions. Phosphorylation was either built in or deleted for the threonine at residue 288 in the second stage of the Solvator tool. Also in this stage, residues 225 and 284 were mutated to cysteines and the MTSL spin label (named CYR1) was added to those residues. C290 was mutated to serine in the unphosphorylated conditions and alanine in the phosphorylated conditions, to match the DEER experimental conditions. After the PDB file was generated, a rectangular solvent box was generated using 10 Å edge distance fit to the protein size, with 300 mM NaCl placed using the Monte-Carlo method.

## Parameterization the MTSL labeled CHARMM simulations

An OpenMM ForceField was instantiated using CHARM36 (*Best et al., 2012*) force field parameters for the protein and water model, along with ADP, TPO, and CYR1 parameter files output by CHARMM-GUI.

## Minimization and equilibration for the MTSL labeled CHARMM simulations

A local energy minimization was performed with no constraints on bonds or angles, which took place in the solvated water box output by CHARMM-GUI and loaded into an OpenMM object. After minimization, 5000 steps of NVT dynamics were run using a Langevin integrator with a time step of 1.0 fs, temperature of 50K and a collision rate of 90.0 $ps^{-1}$. Nonbonded forces were modeled using the particle-mesh Ewald (PME) method with a cutoff distance of 9.0 Å. All other settings remained at default values, except mixed precision was used throughout. After this, a second equilibration was run using 500000 steps of NPT dynamics using a Langevin integrator with a temperature of 300 Kelvin, collision rate of 90 $ps^{-1}$, and timestep of 2.0 fs. A Monte Carlo Barostat was used with pressure of 1 atm and barostat update interval of 50 steps. To minimally relax this structure, 500000 steps of Langevin dynamics were run using a Langevin integrator with a two fs time step, 300 K temperature, and a collision rate of 5 $ps^{-1}$.

## Production simulation for MTSL labeled CHARMM simulations

The resulting system, integrator, and state data from the minimization and equilibration were serialized to XML format for simulation on Folding@Home using a simulation core based on OpenMM 6.3. This was done for all eight conditions, where each combination of starting structure, phosphorylation status and Tpx2 status was set up as a RUN. For each of the RUNs, 100 CLONEs with different initial random velocities and random seeds were simulated on Folding@home, where each clone ran for a maximum of 3 μs (1.5 billion Langevin dynamics steps with all-atom output frames saved every 250,000 steps using mixed precision and a Monte Carlo Barostat with pressure of 1 atm, 300 Kelvin, and barostat frequency of 50). In aggregate, each of the 12 configurations totaled between 75-110μs per starting configuration.

### Data analysis

Distances and torsions were computed using the compute_distances and compute_dihedrals functions in MDTraj v 1.8.0. For the WT Amber simulations, the first 100 ns of each CLONE were discarded. The first 250 ns were discarded for the CHARMM MTSL labeled AurA simulations to allow sufficient relaxation following the introduction of spin probes. Distance probability plots were generated using Seaborn v0.8.1 (https://doi.org/10.5281/zenodo.883859) distplot using the norm_hist parameter. The distance and dihedral contour plots were generated using the kdeplot function in Seaborn v0.8.1. All analysis scripts have been made publically available (*Hubbard et al., 1994*).

# Acknowledgements

We thank LeeAnn Higgins and Todd Markowski (Center for Mass Spectrometry and Proteomics, University of Minnesota) and Joseph Dalluge (Mass Spectrometry Laboratory, University of Minnesota)

for help with mass spectrometry experiments. DEER experiments were performed in the Biophysical Technology Center, University of Minnesota. We thank Tanya Freedman for critical reading of the manuscript, and Wendy Gordon for helpful discussions. This work was supported by NIH grants GM102288, CA217695 (NML), F32GM120817 (EFR) and P30-CA008748 and GM121505 (JDC). JDC, JMB, SMH, and SKA acknowledge support from the Sloan Kettering Institute.

## Additional information

### Competing interests

John D Chodera: is a member of the Scientific Advisory Board for Schrödinger, LLC. The other authors declare that no competing interests exist.

### Funding

| Funder | Grant reference number | Author |
|---|---|---|
| National Institutes of Health | R00 Award GM102288 | Nicholas M Levinson |
| National Institutes of Health | R21 Award CA217695 | Nicholas M Levinson |
| National Institutes of Health | NRSA Award F32GM120817 | Emily F Ruff |
| National Institutes of Health | P30-CA008748 | John D Chodera |
| National Institutes of Health | GM121505 | John D Chodera |

The funders had no role in study design, data collection and interpretation, or the decision to submit the work for publication.

### Author contributions

Emily F Ruff, Conceptualization, Data curation, Formal analysis, Funding acquisition, Investigation, Visualization, Methodology, Writing—original draft; Joseph M Muretta, Software, Formal analysis, Supervision, Validation, Investigation, Methodology, Writing—review and editing; Andrew R Thompson, Software, Formal analysis, Supervision, Investigation, Methodology, Writing—review and editing; Eric W Lake, Software, Investigation, Methodology, Writing—review and editing; Soreen Cyphers, Investigation; Steven K Albanese, Data curation, Software, Formal analysis, Validation, Investigation, Visualization, Methodology, Writing—review and editing; Sonya M Hanson, Resources, Data curation, Software, Formal analysis, Validation, Investigation, Visualization, Methodology, Writing—review and editing; Julie M Behr, Resources, Software, Formal analysis, Investigation, Methodology, Writing—review and editing; David D Thomas, Supervision, Funding acquisition, Validation, Methodology, Writing—review and editing; John D Chodera, Conceptualization, Resources, Data curation, Software, Formal analysis, Supervision, Funding acquisition, Validation, Investigation, Visualization, Methodology, Project administration, Writing—review and editing; Nicholas M Levinson, Conceptualization, Resources, Data curation, Software, Formal analysis, Supervision, Funding acquisition, Validation, Investigation, Visualization, Methodology, Writing—original draft, Project administration, Writing—review and editing

### Author ORCIDs

Steven K Albanese http://orcid.org/0000-0003-0973-5030
Sonya M Hanson http://orcid.org/0000-0001-8960-5353
John D Chodera http://orcid.org/0000-0003-0542-119X
Nicholas M Levinson http://orcid.org/0000-0003-4338-8087

### Decision letter and Author response

Decision letter https://doi.org/10.7554/eLife.32766.018
Author response https://doi.org/10.7554/eLife.32766.019

## Additional files

**Supplementary files**
• Transparent reporting form
DOI: https://doi.org/10.7554/eLife.32766.016

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
