## [Decision Letter]

Thank you for submitting your article "A dynamic mechanism for allosteric activation of Aurora kinase A by activation loop phosphorylation" for consideration by *eLife*. Your article has been reviewed by three peer reviewers, one of whom is a member of our Board of Reviewing Editors, and the evaluation has been overseen by Philip Cole as the Senior Editor. The following individual involved in review of your submission has agreed to reveal his identity: Stefan Knapp (Reviewer #2).

The reviewers have discussed the reviews with one another and the Reviewing Editor has drafted this decision to help you prepare a revised submission.

The reviewers agree that this is a very detailed analysis of the dynamic features of the activation loop of an important protein kinase, and in particular of the DFG motif. It is more comprehensive than any previous study and also introduces new concepts for thinking about this important conserved feature of every protein kinase. The spectroscopic studies are well done and informative and some of the technologies are new. However, the overall organization of the paper is a weakness and the major concepts were not clearly articulated and in some cases appear contradictory. Most importantly, it was felt that the DFG motif cannot be considered independently of the other concerted and correlated motions that are mediated by the C-Helix and by the Tpx2 motif that docks directly onto the C-Helix. It is felt that key issues were lost in the interpretation of the results and in the discussion. To submit a revised manuscript there would need to be major revisions as outlined below. Given the potential importance of these findings we would be willing to consider a substantially revised manuscript.

1) A major concern has to do with the relative roles of Tpx2 and phosphorylation in vivo. You note that Tpx2 operates at the mitotic spindle, where the kinase is kept unphosphorylated by the action of PP6, whereas at the centrosome activation is by phosphorylation at T288 on the activation loop, implying two distinct activation mechanisms. There was no discussion of what Tpx2 actually does in addition to recruiting a phosphatase. Your studies, as well as earlier literature, indicate that Tpx2 and phosphorylation can act synergistically in vitro, but it is not entirely clear whether phosphorylation alone or another factor that serves a role similar to Tpx2 might operate at the centrosome. A revised manuscript would require a clear mechanistic discussion concerning the role of activation loop phosphorylation and Tpx2 docking onto the C-Helix, which are correlated motions.

2) The reviewers feel that the kinetic data in the last part of the paper are not particularly solid. For example, the stopped-flow experiments lead to the model that the DFG-Out configuration is needed for nucleotide release and could be an intermediate step in the catalytic cycle. However, these experiments were performed in the absence of a peptide substrate. What happens under conditions where peptide is present? Overall, the arguments regarding a possible role of the "out" conformation in nucleotide release are not sufficiently substantiated, so we would ask that these portions be dropped from the manuscript unless they are fortified with more extensive measurements with peptide substrate.

3) The activation segment clearly serves to activate kinases, but phosphorylation also provides the docking site for substrates. It is conceivable that binding of a specific substrate will stabilize the DFG-in conformation further and this effect could be dependent on a particular substrate. MYC for instance has been recently described as a substrate that interacts in a fashion that is dependent on the DFG conformation (DFG-out inhibitors displace it). Thus, it is likely that the dynamic properties will change in the presence of specific substrates.

4) Another concern is that ADP and not ATP was used here; this is convenient as ATP will probably lead to additional autophosphorylation during the measurements. However, the γ phosphate, as well as the second metal ion, forms important interactions with the DFG and will almost certainly influence its dynamic properties, especially if the phosphatase is also nearby. ADP concentrations (50-100 uM) are significantly lower than ATP (1-5 mM) concentrations in cells. While AMP-PNP, a non-hydrolyzable ATP analog may not completely mimic ATP, it would be better that ADP. Why was it not used? These issues should be discussed to address the concern that the proposed model is more generally relevant than just to the narrow conditions studied here.

5) The conclusion that the DFG-In population doesn't shift with phosphorylation is based in part on the nitrile-labeled Q185C data in Figure 1. However, Dodson and Bayless, 2012, showed that the labeled Q185C has very little activity without Tpx2. So can they be sure that these data reflect the native enzyme rather than an artifact of the mutation/labeling? The trFRET-derived population data look more convincing in this regard, since they show that labeling has modest effects on activity. On the other hand, with the DEER data, the broader distribution of the phos+ADP vs. unphos +ADP in Figure 3 does suggest a degree of population shift, and includes some species around the 52 Å peak found with Tpx2. What happens if the authors try to fit Gaussians around the major peaks? Overall there are a number of inconsistencies in the different sets of experiments. As indicated above introducing the kinetic data in contrast to the steady state data only adds further confusion.

6) More explicit reference to earlier validation of the spectroscopic techniques, including effects (or lack thereof) on enzyme activity by the introduction of cysteines and probes, use of ADP vs. ATP as a cofactor, etc. should be provided. What new techniques and data are introduced here?

7) To state that regulation of Aurora A by phosphorylation and by Tpx2 are normally mutually exclusive is a simplification that is not substantiated by your data. Without more data and information on the time scales and physiological partners of Aurora A under different conditions, this statement seems to be an over simplification.

8) Another concern is with the final statement in the discussion regarding the evolution of kinases. The evolutionary pressure for kinases is probably not to optimize for catalytic efficiency (i.e. substrate binding and product release) but rather to optimize for regulation via a switch mechanism, which is what you are characterizing here.

[Editors' note: further revisions were requested prior to acceptance, as described below.]

Thank you for resubmitting your work entitled "A dynamic mechanism for allosteric activation of Aurora kinase A by activation loop phosphorylation" for further consideration at *eLife*. Your revised article has been favorably evaluated by Philip Cole (Senior Editor), a Reviewing editor, and two reviewers.

The manuscript has been improved but there are some remaining issues that need to be addressed before acceptance, as outlined below:

The reviewers agree that your revised manuscript is substantially improved, including the addition of new data with AMP-PNP and the removal of the kinetic data, focusing on the dynamic features of the DFG motif and the synergy between the activation loop and the TPX2 docking motif. This is a comprehensive analysis of the dynamic features of these regions that converge at the active site and provides a foundation for others to think about this region. The use of multiple biophysical techniques is a great strength. Although each kinase is regulated in unique ways, all involve the critical coupling of these regions, and these motions are tightly regulated. The results are thus highly relevant to a large set of investigators in the signaling area.

Having said this, the reviewers feel that interpretation of the results need further revision; these do not require additional experiments:

1) The questions addressed in point 3 in the rebuttal letter highlight the peptide substrates and in fact talk about substrates likely diffusing in and out of the active site. These substrates bind weakly (Km 10 mM) so it is not likely that these are physiologically relevant. Indeed, experiments at high peptide concentration indicate no changes on the activation segment conformation. It is important to emphasize that in cells the substrates are not peptides, but proteins that are often tethered to a signaling complex by motifs that are not at the active site. Such substrates can have docking sites that increase their local concentration; if they are tethered by an additional site they are not limited by diffusion once the kinase is activated, even if it is activated in a very transient way. The authors should comment on this, as the whole system may be seen in a different way if one considers it as a tightly assembled signaling complex. In this scenario it is very likely that there could be an as yet unidentified functional equivalent of Tpx2 in the mitotic spindle.

2) The authors note that MYC binds as a pseudosubstrate to AurA. It has, however, also been shown that type II inhibitors displace MYC, thus there is presumably tight coupling between the activation segment and substrate binding. It would be good to discuss this a bit more.

3) The authors argue that the "…C-helix is likely also coupled to these elements, but we note that the scale of the conformational changes of the C-helix in AurA is small (as is the case in the entire AGC lineage due to the constraining influence of the B-helix in these proteins)". This is not evident from crystal structures that show flexibility of the alphaC helix. Admittedly, these structures lack the N-terminus but if the N-terminus constrains alphaC movement this would also pose a problem for the construct used in the spectroscopic data that uses a similar boundary. The authors should attempt to reconcile this issue in the discussion.

4) The synergy between the two regions – the TPX2 docking site (referred to as the hydrophobic motif in other AGC kinases) and the activation loop – is a fascinating conclusion. While this synergy is critical in all kinases and highly regulated, it is not easily trackable in other kinases. Aurora A with its trans mechanism for docking to the hydrophobic motif is an exceptional model system to tease apart this synergy but the results are in general likely to be applicable to other kinases. The authors should at least pose the question of whether phosphorylation of the activation loop actually "drives" the assembly of the active kinase or "facilitates" the assembly of the active kinase.

5) The authors should state clearly that the driving force for evolution of the protein kinases is likely to be regulation and not highest efficiency catalysis.

6) With regard to correlated motions of the C-Helix and the Hydrophobic Spine, we think it is important to include this motif in your analyses as these all seem to be related motions, including the water molecules that you describe. Even though the C-Helix does not substantially move in the AGC kinases, as you point out, it is the alignment of the spine residues that is critical and in the case of the AGC kinases there has been much attention paid to the "capping" of the R-spine with the hydrophobic motif – a process that is highly regulated in unique ways for each AGC kinase. With Aurora A you have a very unique system which allows you to tease apart the allosteric features of the hydrophobic motif as it comes from a different protein. However, the concepts are likely to be similar. It would at least be useful to see one figure that compares the DFG motif and the water molecules, as well as the second Mg ion, with the R-spine in the DFG-In and DFG-out position – even if it is in the supplement.

7) There is a question that relates to the C185 position, which is where you place one of the labels. This residue clearly is packed against the DFG phenylalanine in the DFG-out position. Cys itself is quite hydrophobic as is your probe. You demonstrate that the properties of the modified protein resemble those of the wt protein. However, do you know what happens if you replace this Cys with an Ala? Is this an activating mutation? It would seem that the unique position of this Cys might contribute to maintaining the DFG out inhibited state. Is this a possibility?

---

## [Author Response]

The reviewers agree that this is a very detailed analysis of the dynamic features of the activation loop of an important protein kinase, and in particular of the DFG motif. It is more comprehensive than any previous study and also introduces new concepts for thinking about this important conserved feature of every protein kinase. The spectroscopic studies are well done and informative and some of the technologies are new. However, the overall organization of the paper is a weakness and the major concepts were not clearly articulated and in some cases appear contradictory. Most importantly, it was felt that the DFG motif cannot be considered independently of the other concerted and correlated motions that are mediated by the C-Helix and by the Tpx2 motif that docks directly onto the C-Helix.

We never meant to suggest that the DFG motif can be considered independently of the motions of the activation loop and C-helix. Indeed, we take the striking similarity in the IR and FRET/EPR data – which all show similar nucleotide- and Tpx2-mediated conformational shifts – as a clear indication that conformational changes of different structural elements are tightly coupled across the kinase domain. We concur that the C-helix is likely also coupled to these elements, but we note that the scale of the conformational changes of the C-helix in AurA is small (as is the case in the entire AGC lineage due to the constraining influence of the B-helix in these proteins) and therefore challenging to measure directly, and we have therefore relied on the DFG motif and activation loop as readouts of the overall conformational state of the kinase.

It is felt that key issues were lost in the interpretation of the results and in the discussion. To submit a revised manuscript there would need to be major revisions as outlined below. Given the potential importance of these findings we would be willing to consider a substantially revised manuscript.

We regret that the original manuscript was confusing in places. We believe that the removal of the kinetics section as suggested by the reviewers, and our refocusing of the discussion on the key structural questions, particularly how phosphorylation triggers a structural rearrangement within the DFG-In state, have simplified the revised manuscript and improved the clarity and focus. We believe that the central result of the paper, that phosphorylation alters the structure of the DFG-In state to bring about activation, is now much more prominently and clearly articulated.

1) A major concern has to do with the relative roles of Tpx2 and phosphorylation in vivo. You note that Tpx2 operates at the mitotic spindle, where the kinase is kept unphosphorylated by the action of PP6, whereas at the centrosome activation is by phosphorylation at T288 on the activation loop, implying two distinct activation mechanisms. There was no discussion of what Tpx2 actually does in addition to recruiting a phosphatase. Your studies, as well as earlier literature, indicate that Tpx2 and phosphorylation can act synergistically in vitro, but it is not entirely clear whether phosphorylation alone or another factor that serves a role similar to Tpx2 might operate at the centrosome. A revised manuscript would require a clear mechanistic discussion concerning the role of activation loop phosphorylation and Tpx2 docking onto the C-Helix, which are correlated motions.

The reviewers raise a good point. Our assertion that the roles of Tpx2 and phosphorylation were largely independent in vivo was based on considerable previous work by other labs. For instance, the work of Toya et al., 2011 showed that spindle-associated AurA is not phosphorylated on T288, and that a T288A mutant of AurA does not block the spindle functions. Further, work by Zeng et al., 2010 has confirmed that the major population of AurA on the spindle is not phosphorylated on T288 due to the action of the PP6 phosphatase. Similarly, studies of Tpx2 localization and knockdowns (Kufer et al., 2002) showed that Tpx2 is not present at the centrosome, does not interact with centrosomal AurA and is not required for its centrosomal localization and function. These in vivo studies formed the basis for our assertion that the two activation mechanisms are independent. We have edited the Introduction section to more clearly describe these points.

We concur with the reviewers that our biochemical data showing striking synergy between Tpx2 and phosphorylation could be considered puzzling in light of the in vivo work, and might potentially indicate the existence of a small but functionally-important cellular pool of AurA that is activated by both Tpx2 and phosphorylation. In light of this we have toned down the assertion of the independence of the Tpx2 and phosphorylation mechanisms in the revised Discussion section. However, we note that the presence of both synergy and independence in the actions of Tpx2 and phosphorylation was previously reported by Dodson and Bayliss, 2012 (“Activation of Aurora-A kinase by protein partner binding and phosphorylation are independent and synergistic”). We now include a new section at the end of the Results that describes how our model for a novel autoinhibited DFG-In substate provides an explanation for the synergy: disruption of the autoinhibited state by phosphorylation promotes the assembly of the Tpx2 binding site.

Finally, we have further amended the Discussion section of the manuscript to make the following important connection with the related AGC family kinases and our previous work, which provides some context for considering the relative synergism versus independence of phosphorylation and Tpx2. Our previous work (Cyphers et al., 2017) showed that the ability of AurA to be strongly activated by either phosphorylation alone or by Tpx2 alone (at least in vitro) was due to an enhanced regulatory spine in the Aurora kinase lineage involving a unique polar residue that coordinates a water-mediated hydrogen bond network in the active site. We showed that removal of this feature by mutagenesis, essentially converting AurA into an AGC-like kinase, prevented individual activation of AurA by phosphorylation or Tpx2 alone, such that both were required simultaneously for activation. This is analogous to the situation found in the AGC kinases where hydrophobic motif engagement and phosphorylation occur as a concerted activation process. We believe it likely that the emergence of the unique hydrogen bond network in AurA was necessary to allow for independent activation of spindle- and centrosome-associated AurA pools. We have now included a discussion of this point in the revised manuscript.

We certainly concur with the reviewers that the existence of a Tpx2-like element that further modulates AurA activity at the centrosome is possible; while identification of this is outside of the scope of this work, we have mentioned the possibility of further modulation of centrosomal AurA by protein-protein interactions in the revised Discussion section.

2) The reviewers feel that the kinetic data in the last part of the paper are not particularly solid. For example, the stopped-flow experiments lead to the model that the DFG-Out configuration is needed for nucleotide release and could be an intermediate step in the catalytic cycle. However, these experiments were performed in the absence of a peptide substrate. What happens under conditions where peptide is present? Overall, the arguments regarding a possible role of the "out" conformation in nucleotide release are not sufficiently substantiated, so we would ask that these portions be dropped from the manuscript unless they are fortified with more extensive measurements with peptide substrate.

We agree with the reviewers that the kinetic data were a weakness of the original paper. Although we maintain that the question of the functional role of the large DFG-Out subpopulation of phosphorylated AurA is an important and interesting question, we concur that this is outside the scope of the current work, and that the inclusion of the kinetic data unnecessarily complicated the manuscript and led to confusion. We have therefore decided to remove the kinetic section and refocus the manuscript on the key structural effects of phosphorylation.

3) The activation segment clearly serves to activate kinases, but phosphorylation also provides the docking site for substrates. It is conceivable that binding of a specific substrate will stabilize the DFG-in conformation further and this effect could be dependent on a particular substrate. MYC for instance has been recently described as a substrate that interacts in a fashion that is dependent on the DFG conformation (DFG-out inhibitors displace it). Thus, it is likely that the dynamic properties will change in the presence of specific substrates.

The reviewers bring up an excellent point about the potential role of substrates in modulating AurA conformation. We performed several experiments in an attempt to dissect the effects of substrate binding on the conformational dynamics of AurA. Unfortunately, the available consensus substrate peptides for AurA are relatively poor, with measured Km values in the millimolar range. We in fact performed IR, FRET and EPR experiments in the presence of 10 mM of the substrate peptide kemptide, which conforms to the consensus sequence determined for AurA by Ferrari et al., 2005, and found no effect on the conformation of the kinase using any of these methods. We have included the EPR data in the revised paper (Figure 4—figure supplement 2). We think it likely that substrate recognition is in fact transient, with the substrate diffusing into the active site and interacting weakly and briefly with the kinase.

We note that MYCN actually binds as a pseudosubstrate to AurA, and although it binds tightly and does indeed trap the kinase in the active state, as shown by the work of Richards et al., 2016, it is important to appreciate that this binding is a non-catalytic scaffolding function of AurA, and does not reflect a normal kinase:substrate interaction.

4) Another concern is that ADP and not ATP was used here; this is convenient as ATP will probably lead to additional autophosphorylation during the measurements. However, the γ phosphate, as well as the second metal ion, forms important interactions with the DFG and will almost certainly influence its dynamic properties, especially if the phosphatase is also nearby. ADP concentrations (50-100 uM) are significantly lower than ATP (1-5 mM) concentrations in cells. While AMP-PNP, a non-hydrolyzable ATP analog may not completely mimic ATP, it would be better that ADP. Why was it not used? These issues should be discussed to address the concern that the proposed model is more generally relevant than just to the narrow conditions studied here.

We did include steady-state FRET experiments with AMPPNP in the original manuscript, and observed very similar results to those with ADP. However, given the importance of this question we agreed with the reviewers that we needed to repeat the other experiments with AMPPNP as well. We have now repeated both the time-resolved FRET and DEER experiments with AMPPNP and observe very similar results to those obtained with ADP (Figure 2 and Figure 4—figure supplement 2). In particular, the DEER experiments with AMPPNP nicely confirm that nucleotide alone does not push either phosphorylated or unphosphorylated AurA fully into the DFG-In state, and that Tpx2 is required to do this in both cases (Figure 4—figure supplement 2). This comparison between unphosphorylated and phosphorylated AurA bound to AMPPNP further highlights that phosphorylation does lead to a slight increase in the sampling of spin-spin distances beyond 50 angstroms, as pointed out by the reviewers in point 5 below. As discussed in response below, this result is consistent with our model that the structure of the DFG-In state is altered by phosphorylation such that it contributes a spin-spin distance of around 52 angstroms to the distance distribution, instead of a distance of around 49 angstroms in the absence of phosphorylation.

Although it might seem surprising that coordination of Mg-ATP by the DFG Aspartate does not trap the kinase in the DFG-In state, it arguably makes sense from the standpoint of using the DFG equilibrium for regulatory control, as Tpx2 does. If ATP binding was tightly coupled to the DFG equilibrium, then the high concentration of ATP in the cell would force AurA into the DFG-In state, overriding any ability of Tpx2 or other factors to influence the DFG equilibrium. Instead, nature has ensured that nucleotide is only weakly coupled to the DFG equilibrium, so that binding and release can occur without overriding the regulatory status of the enzyme.

5) The conclusion that the DFG-In population doesn't shift with phosphorylation is based in part on the nitrile-labeled Q185C data in Figure 1. However, Dodson and Bayliss, 2012, showed that the labeled Q185C has very little activity without Tpx2. So can they be sure that these data reflect the native enzyme rather than an artifact of the mutation/labeling? The trFRET-derived population data look more convincing in this regard, since they show that labeling has modest effects on activity. On the other hand, with the DEER data, the broader distribution of the phos+ADP vs. unphos +ADP in Figure 3 does suggest a degree of population shift, and includes some species around the 52 Å peak found with Tpx2. What happens if the authors try to fit Gaussians around the major peaks? Overall there are a number of inconsistencies in the different sets of experiments. As indicated above introducing the kinetic data in contrast to the steady state data only adds further confusion.

The reviewers are absolutely correct that the kinase activity of the nitrile-modified enzyme is minimal without Tpx2, and that this could in principle be a sign of a deleterious effect of the labeling on the conformational equilibrium. Indeed, we would not seek to conclude solely from the IR data that the DFG equilibrium is not shifted by phosphorylation. However, the conformation of the DFG motif is tightly coupled to that of the activation loop, as is established from numerous x-ray structures of kinases as well as our results with Tpx2, and the TR-FRET and DEER data do both support the interpretation that phosphorylation has minimal effect on the conformational equilibrium of the activation loop. Indeed, the slight difference between the DEER distributions of AurA:ADP with and without phosphorylation mentioned by the reviewers (see Figure 4 of the revised manuscript) can actually be attributed to the different structures of the DFG-In state in the two cases, as is nicely demonstrated by our experiments with SNS-314 (Figure 4): the DFG-in state of phosphorylated AurA contributes a spin-spin distance of ~52-53 angstroms to the distance distribution, whereas the DFG-in state of unphosphorylated AurA instead contributes a 49-Å distance. The new experiments with AMPPNP included in the revised manuscript further support this result (see Figure 4—figure supplement 2, where the darker blue shading highlights the increased sampling of distances beyond 50 angstroms in the phosphorylated sample). We regret that this was not clear in the original manuscript, and in the revised manuscript we have consolidated the EPR data into a single figure and heavily edited the discussion of the data to help highlight this critical point. We have also further verified the phosphorylation-driven structural change within the DFG-In state using a separate set of DEER experiments in which one of the spin-labels was moved to S283 (instead of S284) and SNS-314 again used to isolate the DFG-In state. These data confirm that phosphorylation causes a pronounced structural change within the DFG-In state (Figure 4—figure supplement 3).

We agree that the inclusion of the kinetic data led to confusion and have removed this section from the revised manuscript and refocused the discussion on how phosphorylation alters the structure of the DFG-in state of AurA to activate the kinase, the central conclusion of this work.

We did attempt to fit the distance distributions to Gaussian functions as suggested by the reviewers to estimate the degree to which the DFG-Out and DFG-In states are populated in each sample. However, because of the complexity of the Tikhonov-derived distance distributions, with three to four distinct peaks contributed by each structural state due to alternative rotamers of the spin probes, this fitting procedure is problematic and we did not find the results to be very reliable or informative. We decided to focus instead on more qualitative descriptions of the distributions which nonetheless serve to illustrate the central points that a) the conformational ensemble of phosphorylated AurA is heterogeneous in the absence of Tpx2 and b) phosphorylation alters the structure of the DFG-In state as reflected in the contribution of a 52-Å distance to the distribution.

We feel that in the revised manuscript the apparent inconsistencies between the different methods have been resolved. Specifically, all three experimental methods strongly support the key observations that:

A) Unphosphorylated and phosphorylated AurA reside in a similar conformational equilibrium between DFG-Out and DFG-In states.

B) Tpx2 binding is required for both unphosphorylated and phosphorylated AurA to homogeneously adopt the DFG-In state.

6) More explicit reference to earlier validation of the spectroscopic techniques, including effects (or lack thereof) on enzyme activity by the introduction of cysteines and probes, use of ADP vs. ATP as a cofactor, etc. should be provided. What new techniques and data are introduced here?

We have edited the manuscript to ensure that the validation data are clearly presented and previous work appropriately cited throughout. We have now included additional information on the preparation of the phosphorylated Q185CN form of AurA for IR in Figure 1—figure supplement 1, and in the interest of full disclosure have added a sentence to the figure legend highlighting the substantially decreased kinase activity of this sample observed in the absence of Tpx2, which is also shown in the figure. We feel that the fact that the incorporation of fluorescent dyes and spin-labels has very little effect on kinase activity is a major strength of the analysis that bolsters the validity of the story. An additional important point is the considerable validation of the Cys-lite construct of AurA performed by Richard Bayliss’ group, including validation of kinase activity (Rowan et al., 2013) and the dissection of how the C290A mutation simplifies the phosphorylation pattern obtained during recombinant expression in *E. coli*, and the crystal structure of the Cys-lite construct determined by Burgess et al., 2015 which was essentially indistinguishable from the WT. We have now cited these advances in the Materials and methods section.

7) To state that regulation of Aurora A by phosphorylation and by Tpx2 are normally mutually exclusive is a simplification that is not substantiated by your data. Without more data and information on the time scales and physiological partners of Aurora A under different conditions, this statement seems to be an over simplification.

As discussed above in response to point 1, we concur that we may have overstated this point, which was based on the in vivo work of other groups and not on our own data, which indeed could be seen to potentially support a physiological role for Tpx2 and phosphorylation acting together. We have edited the manuscript to tone down this assertion and removed the “mutually exclusive” phrase. We have also included in the revised Discussion section a comparison between the Tpx2/phosphorylation mechanisms of AurA and the activation mechanisms of the related AGC family kinases. We feel this discussion helps put our work in context and more accurately describes the dual activation mechanisms of AurA as a variation on a theme rather than an entirely unique property.

8) Another concern is with the final statement in the discussion regarding the evolution of kinases. The evolutionary pressure for kinases is probably not to optimize for catalytic efficiency (i.e. substrate binding and product release) but rather to optimize for regulation via a switch mechanism, which is what you are characterizing here.

We agree with the reviewers that the primary selective pressure operating on kinases is regulatory, and in the light of the removal of the kinetics section and the refocusing of the manuscript on the structural story, we have amended this last section of the Discussion. We do think that the potential function of the DFG-Out subpopulation is an intriguing question, but agree that this should be left for a future analysis.

[Editors' note: further revisions were requested prior to acceptance, as described below.]

1) The questions addressed in point 3 in the rebuttal letter highlight the peptide substrates and in fact talk about substrates likely diffusing in and out of the active site. These substrates bind weakly (Km 10 mM) so it is not likely that these are physiologically relevant. Indeed, experiments at high peptide concentration indicate no changes on the activation segment conformation. It is important to emphasize that in cells the substrates are not peptides, but proteins that are often tethered to a signaling complex by motifs that are not at the active site. Such substrates can have docking sites that increase their local concentration; if they are tethered by an additional site they are not limited by diffusion once the kinase is activated, even if it is activated in a very transient way. The authors should comment on this, as the whole system may be seen in a different way if one considers it as a tightly assembled signaling complex. In this scenario it is very likely that there could be an as yet unidentified functional equivalent of Tpx2 in the mitotic spindle.

This is an excellent point and we have amended the Discussion section to include a sentence pointing out how complex formation and substrate docking in vivo may further modulate the dynamics of the phosphorylated enzyme.

2) The authors note that MYC binds as a pseudosubstrate to AurA. It has, however, also been shown that type II inhibitors displace MYC, thus there is presumably tight coupling between the activation segment and substrate binding. It would be good to discuss this a bit more.

We have included a mention of the NMYC pseudo substrate interaction and how it modulates the kinase dynamics in the revised Discussion section, in the same section that deals with the broader issue of substrate docking raised in point 1.

3) The authors argue that the "…C-helix is likely also coupled to these elements, but we note that the scale of the conformational changes of the C-helix in AurA is small (as is the case in the entire AGC lineage due to the constraining influence of the B-helix in these proteins)". This is not evident from crystal structures that show flexibility of the alphaC helix. Admittedly, these structures lack the N-terminus but if the N-terminus constrains alphaC movement this would also pose a problem for the construct used in the spectroscopic data that uses a similar boundary. The authors should attempt to reconcile this issue in the discussion.

We have amended the Discussion section of the manuscript to further discuss the role of the aC-helix and how it is coupled to phosphorylation and Tpx2 binding.

To be clear we did not intend to suggest above that the aC-helix doesn’t move at all in AurA, but rather to draw a clear distinction between the dynamics in AurA/AGC kinases versus in the classes of kinases like the CDKs and Src family in which the aC-helix undergoes a large-scale movement into an alternative, stable, and heavily-populated autoinhibitory state. The latter mode of motion is inhibited by the B-helix in AurA/AGC kinases. The reviewers are completely correct that the aC-helix may still be “dynamic” in AurA/AGC kinases, as evidenced by disorder in crystal structures. Indeed we detected these dynamics in our earlier molecular dynamics work (Cyphers et al., 2017) and found that they were suppressed by Tpx2. The smaller magnitude of the motions characteristic of these latter “dynamics” simply make it difficult to study them experimentally by e.g. DEER spectroscopy, which is why we have not measured these dynamics experimentally. This would be an interesting topic for a future work but is clearly outside the scope of the current study.

4) The synergy between the two regions – the TPX2 docking site (referred to as the hydrophobic motif in other AGC kinases) and the activation loop – is a fascinating conclusion. While this synergy is critical in all kinases and highly regulated, it is not easily trackable in other kinases. Aurora A with its trans mechanism for docking to the hydrophobic motif is an exceptional model system to tease apart this synergy but the results are in general likely to be applicable to other kinases. The authors should at least pose the question of whether phosphorylation of the activation loop actually "drives" the assembly of the active kinase or "facilitates" the assembly of the active kinase.

We believe that our data, particularly the EPR results, show that phosphorylation really drives the assembly of the active state, albeit without stabilizing the active state relative to the DFG-Out state, which is in and of itself remarkable. The way we prefer to think about it is that the data argue that phosphorylation acts by altering the detailed shape of the conformational energy landscape for the DFG-In state, rather than by changing the depth of the free energy well (with respect to that for the DFG-Out state). We have amended the wording in the Discussion to clarify this point.

5) The authors should state clearly that the driving force for evolution of the protein kinases is likely to be regulation and not highest efficiency catalysis.

We have amended the Discussion to make this point explicit.

6) With regard to correlated motions of the C-Helix and the Hydrophobic Spine, we think it is important to include this motif in your analyses as these all seem to be related motions, including the water molecules that you describe. Even though the C-Helix does not substantially move in the AGC kinases, as you point out, it is the alignment of the spine residues that is critical and in the case of the AGC kinases there has been much attention paid to the "capping" of the R-spine with the hydrophobic motif – a process that is highly regulated in unique ways for each AGC kinase. With Aurora A you have a very unique system which allows you to tease apart the allosteric features of the hydrophobic motif as it comes from a different protein. However, the concepts are likely to be similar. It would at least be useful to see one figure that compares the DFG motif and the water molecules, as well as the second Mg ion, with the R-spine in the DFG-In and DFG-out position – even if it is in the supplement.

We have added a new figure to the supplement (Figure 1—figure supplement 2) showing a comparison of the R-spine and DFG motif in the DFG-In and DFG-Out states. One of the interesting features of the DFG-Out state is that the role of the DFG phenylalanine in the R-spine is replaced by the DFGx tryptophan residue (W277), which partially restores the hydrophobic contacts of the R-spine. The polar interactions of Q185 with the water molecules characteristic of the active state, are, however, lost. We have also amended the Discussion section to explicitly mention our earlier result (Cyphers et al) that Tpx2 stabilizes the C-helix and regulatory spine.

7) There is a question that relates to the C185 position, which is where you place one of the labels. This residue clearly is packed against the DFG phenylalanine in the DFG-out position. Cys itself is quite hydrophobic as is your probe. You demonstrate that the properties of the modified protein resemble those of the wt protein. However, do you know what happens if you replace this Cys with an Ala? Is this an activating mutation? It would seem that the unique position of this Cys might contribute to maintaining the DFG out inhibited state. Is this a possibility?

In our earlier work (see Cyphers et al., 2017) we replaced the Q185 residue with several different amino acids found at that position in different ePKs. In total we tested Q185C, Q185N, Q185L, Q185M and Q185H. We found that all mutations were deleterious to the activity of the phosphorylated enzyme, but that the activity of most of the mutants could be largely or completely rescued by activating with both phosphorylation and Tpx2. Interestingly, Q185N was the most deleterious mutation, and could only be partially rescued. We did not test the Q185A mutation, as alanine is not commonly found at this position in ePKs. It is certainly possible that some of the effects of the substitutions on kinase activity are due to differentially stabilizing the DFG-Out and DFG-In states, but as described in our Nat Chem Biol paper, we attributed most of the effects to the disruption of the water-mediated hydrogen bond network nucleated by Q185. Given the generally good agreement between the IR data, obtained with the unnatural Q185CN probe, and the FRET and EPR data, obtained with a native Q185 residue in the active site, it seems likely that any perturbation of the DFG equilibrium by the probe is relatively minor, although we can’t rule out some effect. We have added a sentence to the IR section pointing out this caveat.